# Establishing synthetic ribbon-type active zones in a heterologous expression system

Rohan Kapoor[1,2,3,4], Thanh Thao Do[5], Niko Schwenzer[4,6,7], Arsen Petrovic[4,5], Thomas Dresbach[8], Stephan E Lehnart[4,6,7], Rubén Fernández-Busnadiego[4,5], Tobias Moser[1,2,3,4]*

[1]Institute for Auditory Neuroscience and InnerEarLab, University Medical Center Göttingen, Göttingen, Germany; [2]Auditory Neuroscience and Synaptic Nanophysiology Group, Max Planck Institute for Multidisciplinary Sciences, Göttingen, Germany; [3]IMPRS Molecular Biology, Göttingen Graduate School for Neuroscience and Molecular Biosciences, University of Göttingen, Göttingen, Germany; [4]Cluster of Excellence 'Multiscale Bioimaging: from Molecular Machines to Networks of Excitable Cells' (MBExC 2067), University of Göttingen, Göttingen, Germany; [5]Institute for Neuropathology, University Medical Centre Göttingen, Göttingen, Germany; [6]Cellular Biophysics & Translational Cardiology Section, Heart Research Center Göttingen, University Medical Center Göttingen, Göttingen, Germany; [7]Department of Cardiology and Pneumology, University Medical Center Göttingen, Göttingen, Germany; [8]Institute of Anatomy and Embryology, University Medical Center Göttingen, Göttingen, Germany

*For correspondence:
tmoser@gwdg.de

Competing interest: The authors declare that no competing interests exist.

## eLife Assessment

The authors take a synthetic approach by introducing synaptic ribbon proteins into HEK cells to analyze how these assemblies cluster calcium channels at the active zone. Using a synapse-naive heterologous expression system and overexpression-based strategy is **valuable**, as it establishes a promising model for studying molecular interactions at the active zone. The study is built on a **solid** combination of super-resolution microscopy and electrophysiology, though it currently falls short of replicating the full functional properties of native ribbon synapses and instead resembles a multiprotein complex that partially mimics ribbon-type active zones.

**Abstract** Encoding of several sensory modalities into neural signals is mediated by ribbon synapses. The synaptic ribbon tethers synaptic vesicles at the presynaptic active zone (AZ) and may act as a super-scaffold organizing AZ topography. Here, we employed a synthetic biology approach to reconstitute structures mimicking ribbon-type AZs in human embryonic kidney 293 (HEK293) cells for probing minimal molecular requirements and studying presynaptic $Ca^{2+}$ channel clustering. Co-expressing a membrane-targeted version of the AZ protein Bassoon and the ribbon core protein RIBEYE, we observed structures recapitulating basic aspects of ribbon-type AZs, which we call *synthetic* ribbons or *SyRibbons*. Super-resolution stimulated emission depletion (STED) microscopy and cryo-correlative electron tomography revealed *SyRibbons* were similar to native ribbons at AZs of cochlear inner hair cells in shape and size. *SyRibbons* with $Ca^{2+}$ channel clusters formed upon additional expression of $Ca_V1.3$ $Ca^{2+}$ channels and RIM-binding protein 2 (RBP2). $Ca_V1.3$ $Ca^{2+}$ channel clusters associated with *SyRibbons* were larger than ribbonless $Ca_V1.3$ $Ca^{2+}$ channel clusters, and functional analysis by $Ca^{2+}$ imaging in combination with patch clamp showed partial confinement

of the $Ca^{2+}$ signal at *SyRibbons*. In summary, we identify $Ca^{2+}$ channels, RBP, membrane-anchored Bassoon, and RIBEYE as minimal components for reconstituting a basic ribbon-type AZ. *SyRibbons* might complement animal studies on molecular interactions of AZ proteins.

## Introduction

Synapses of sensory receptor cells of the inner ear and the retina are hallmarked by the presence of an electron-dense structure called the synaptic ribbon, which tethers synaptic vesicles (SVs) at the presynaptic active zone (AZ; *Moser et al., 2019*; *Matthews and Fuchs, 2010*). Proposed functional roles of the ribbon include (i) its role as an SV replenishment machine for tireless neurotransmission (*Bunt, 1971*; *Holt et al., 2004*; *Khimich et al., 2005*; *Frank et al., 2010*; *Snellman et al., 2011*; *Graydon et al., 2011*; *Vaithianathan et al., 2016*), (ii) 'super-scaffold' regulating the abundance, topography, and function of AZ players, such $Ca^{2+}$ channels and release sites, and their tight coupling (*Khimich et al., 2005*; *Frank et al., 2010*; *Wong et al., 2014*; *Maxeiner et al., 2016*; *Jean et al., 2018*; *Grabner and Moser, 2021*), and (iii) coordination of SV release (*Heidelberger et al., 1994*; *Glowatzki and Fuchs, 2002*; *Singer et al., 2004*; *Edmonds et al., 2004*; *Mehta et al., 2013*). Yet, our molecular understanding of how these postulated functions are executed by synaptic ribbons is still limited.

Likewise, the clustering of $Ca^{2+}$ channels at the AZ, how it varies among different or even the same synapse types, and how the ribbon contributes toward this remain active areas of research (*Wichmann and Kuner, 2022*; *Moser et al., 2023*). For example, ribbon synapses of mature inner hair cells (IHCs) assemble $Ca_V1.3$ voltage-gated $Ca^{2+}$ channels at the base of the ribbon, but vary in shape, size, and channel complement of the $Ca_V1.3$ clusters (*Frank et al., 2010*; *Wong et al., 2014*; *Neef et al., 2018*) partially dependent on the position within the IHC (*Ohn et al., 2016*; *Özçete and Moser, 2021*). IHC synapses lacking AZ-anchored ribbons in Bassoon mutant mice showed fewer $Ca^{2+}$ channels with altered topography (*Frank et al., 2010*; *Neef et al., 2018*). While genetic disruption of $Ca^{2+}$ channel-tethering AZ proteins RIM2 (*Jung et al., 2015*) and RBP2 (*Krinner et al., 2017*) reduced the abundance of $Ca^{2+}$ channels at IHC synapses, their topography retained the typical stripe-like shape of $Ca^{2+}$ channel clusters. Constitutive deletion of RIBEYE, the core scaffold protein of the synaptic ribbon, provided so far the most direct test of the scaffolding role of the ribbon (*Maxeiner et al., 2016*; *Jean et al., 2018*; *Becker et al., 2018*; *Grabner and Moser, 2021*). Here, ribbon-less synapses of IHCs showed an altered $Ca^{2+}$ channel topography with a more spatially widespread presynaptic $Ca^{2+}$ signal but intact $Ca^{2+}$ channel complement (*Jean et al., 2018*). Moreover, the voltage dependence of activation, as well as the inactivation of $Ca^{2+}$ channels, appeared to be altered, suggesting a possible role of the synaptic ribbon in regulating $Ca^{2+}$ channel physiology (*Jean et al., 2018*). Changes in $Ca^{2+}$ channel physiology upon deletion of RIBEYE were also found for $Ca_V1.4$ at ribbon-less rod photoreceptor AZs (*Grabner and Moser, 2021*). Finally, IHCs lacking piccolino, a ribbon-synapse-specific isoform of the AZ protein piccolo, show AZs with smaller synaptic ribbons and altered clustering, yet normal complement of $Ca^{2+}$ channels (*Michanski et al., 2023*).

So how does the synaptic ribbon tune into the orchestra of AZ proteins that collectively organize $Ca^{2+}$ channel abundance, topography, and function? To what extent can the effects observed upon disruption of Bassoon be attributed to the concomitant loss of the ribbon? These questions seem even more relevant given that, to our knowledge, no direct interactions between Bassoon (*Frank et al., 2010*) or RIBEYE and $Ca_V1.3$ $Ca^{2+}$ channel have been described so far. Here, we adopted a bottom-up synthetic biology approach to assemble a minimal presynaptic protein machinery required to bring together RIBEYE and $Ca^{2+}$ channels in a 'synapse-naïve' expression system to assess the role of the synaptic ribbon in regulating $Ca^{2+}$ channel clustering and physiology. Similar approaches have previously been employed for reconstituting aspects of conventional synapses, for instance, in 'hemi-synapses' between co-cultured neurons and non-neuronal cells overexpressing presynaptic or postsynaptic components (for review, see *Craig et al., 2006*). More recently, Munc13-1 supramolecular assemblies which recruit release machinery proteins were reconstituted (*Sakamoto et al., 2018*).

Expression of RIBEYE in synapse-naïve cell lines such as COS-7 cells led to large cytosolic assemblies (*Schmitz et al., 2000*). Ribbon-like electron-dense structures with vesicles have also been reported upon heterologous RIBEYE expression in retinal progenitor cell line (R28; *Magupalli et al., 2008*). However, these cells might not be considered synapse-naïve, and the presence and function of $Ca^{2+}$

channels at synthetic ribbon synapses remained to be studied. The need for co-expressing at least three $Ca^{2+}$ channel subunits ($Ca_V\alpha_x$, $Ca_V\beta_x$, and $Ca_V\alpha_2\delta_x$) in synapse-naïve cell lines renders acute co-expression of AZ multidomain proteins cumbersome. Here, we took advantage of synapse-naïve human embryonic kidney 293 (HEK293) cells stably expressing inducible $Ca_V1.3\alpha1$, which is the predominant subtype of voltage-gated $Ca^{2+}$ channels at IHC ribbon synapses (*Brandt et al., 2003*; *Platzer et al., 2000*; *Dou et al., 2004*), as well as constitutively expressed $Ca_V\beta_3$ and $Ca_V\alpha_2\delta_1$. We co-expressed RIBEYE along with membrane-targeted Bassoon and observed structures with striking resemblance to IHC synaptic ribbon-type AZs nicknamed '*SyRibbons*'. We characterized the structure and function of these synthetic ribbon-type AZs and identified Bassoon, RBP2, RIBEYE, and $Ca_V1.3$ channels as the minimal components required for assembling a basic ribbon-type AZ. We observed larger $Ca_V1.3$ channel clusters near *SyRibbons* and with partially localized $Ca^{2+}$ influx upon stimulation. Our results support the role of synaptic ribbons in promoting the formation of $Ca_V1.3$ channel micro-clustering. Synthetic ribbon-type AZs offer a novel approach for functionally studying protein-protein interaction and will likely complement, refine, and reduce experiments on native ribbon synapses.

## Results

### Membrane-targeted Bassoon recruits RIBEYE to the cell membrane in HEK293 cells where structures mimicking IHC synaptic ribbons are formed

HEK293 cells have the advantage of not expressing the synaptic machinery components studied here. This provides a clean background for reconstituting synthetic synapses from a minimal set of

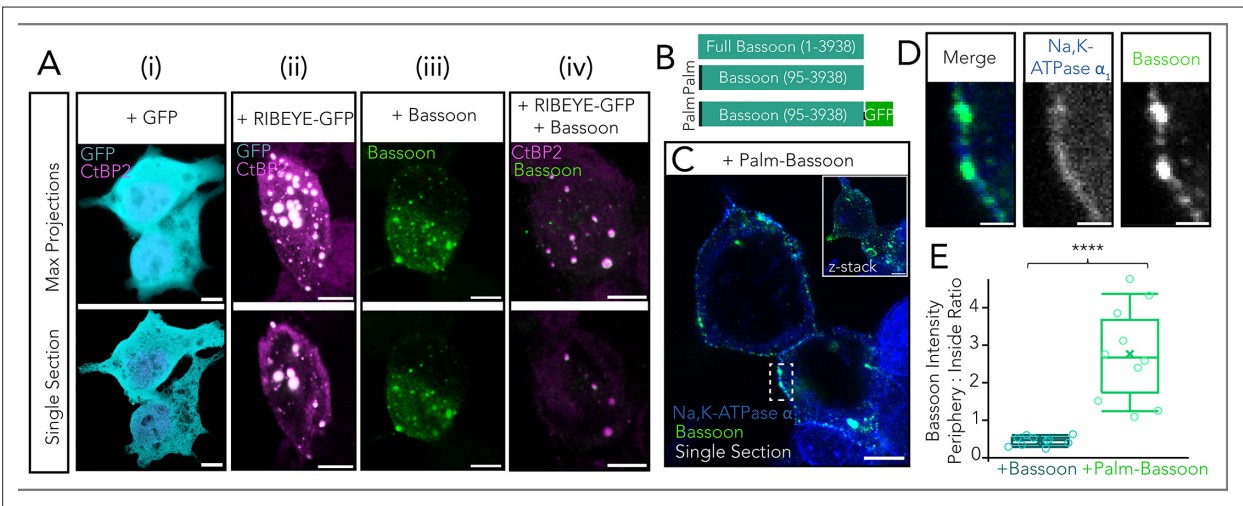

**Figure 1.** Membrane targeting of Bassoon using a palmitoylation consensus sequence. (**A**) Representative confocal images of human embryonic kidney 293 (HEK293) cells transfected with (i) GFP only (CtBP2/RIBEYE in magenta labeling nuclei, GFP in cyan), (ii) RIBEYE-GFP (CtBP2/RIBEYE in magenta, GFP in cyan), (iii) Bassoon (green), and (iv) RIBEYE-GFP and Bassoon. Note that antibodies against RIBEYE B-domain and the nuclear transcription factor CtBP2 result in similar staining patterns as CtBP2 is transcribed from the same gene as RIBEYE and is identical to the RIBEYE-B domain except for the first 20 N-terminal amino acids. The upper panel shows maximum projections, and the lower panel shows exemplary single sections from confocal stacks. Scale bar = 5 µm. (**B**) Schematic of construct for membrane targeting of Bassoon. The first 95 amino acids from full-length Bassoon were replaced with a palmitoylation consensus sequence from GAP43. Constructs without and with a C-terminal GFP tag were used, as depicted. (**C**) Sample confocal image (single section) showing membrane-targeted palm-Bassoon (green) expressed in HEK293 cells appears as puncta distributed along the periphery of the cell, marked by Na, K-ATPase $\alpha_1$ (blue). The inset shows a maximal projection of the confocal section. Scale bar = 5 µm. (**D**) Zoom-in from (**C**) shows colocalization of palm-Bassoon with membrane marker Na, K-ATPase $\alpha_1$. Scale bar = 1 µm. Schematic of construct for membrane targeting of Bassoon. (**E**) Quantification of Bassoon signal intensity at periphery vs inside of cell. Cells expressing palm Bassoon (N = 10 cells) clearly show a higher peripheral distribution compared to cells expressing full-length Bassoon (N = 10 cells), ****p<0.0001, Mann-Whitney-Wilcoxon test. Overlaid data points represent individual cells, crosses represent the mean values, central band indicates the median, whiskers represent 90/10 percentiles, and boxes represent 75/25 percentiles.

The online version of this article includes the following figure supplement(s) for figure 1:

**Figure supplement 1.** Subcellular distribution of RIBEYE and palm-Bassoon clusters in human embryonic kidney 293 (HEK293) cells.

synaptic proteins. Our first step toward assembling a ribbon-type AZ in a heterologous expression system was to express RIBEYE, the core scaffold protein of the synaptic ribbon, and target it to the cell membrane. We performed transient transfection of a RIBEYE construct with a C-terminal EGFP tag in HEK293 cells and observed large spherical clusters of RIBEYE that appeared largely cytosolic (*Figure 1Aii*), in contrast to a diffuse cytosolic distribution when merely expressing EGFP (*Figure 1Ai*). These RIBEYE clusters form due to self-assembling properties of RIBEYE via multiple sites of homophilic interaction, as have been demonstrated before in several cell lines. They do not colocalize with the endoplasmic reticulum (ER), Golgi apparatus, or lysosomes and, hence, appear unlikely to reflect RIBEYE entrapped in protein trafficking pathways or to represent degradation products of overexpressed RIBEYE (*Figure 1—figure supplement 1A, B, and C*).

Next, for membrane targeting of these cytosolic RIBEYE clusters, we co-expressed the multidomain, cytomatrix of the AZ protein Bassoon (*Figure 1Aiii*). Prior work on the molecular underpinnings of ribbon synapses had identified Bassoon (*tom Dieck et al., 1998*) to critically contribute to anchoring the synaptic ribbon to the AZ membrane (*Khimich et al., 2005*; *Dick et al., 2003*; *tom Dieck et al., 2005*). Co-expression of full-length Bassoon along with RIBEYE in HEK293 cells showed colocalizing clusters of the two proteins that, however, remained largely cytosolic (*Figure 1Aiv*).

Next, for plasma membrane targeting of Bassoon, we generated a construct by removing the first 95 N-terminal amino acids of Bassoon and replacing these with a palmitoylation consensus sequence of the neuronal growth-associated protein 43 (GAP43). We refer to this as 'palm-Bassoon' throughout, and we used constructs with and without a C-terminal EGFP tag (*Figure 1B*). Expression of either of these palm-Bassoon constructs in HEK293 cells showed comparable immunofluorescence patterns with Bassoon puncta spread across the periphery of the cell, largely colocalizing with the endogenously expressed plasma membrane-standing Na, K-ATPase $\alpha_1$ (data representative of 3 transfections, *Figure 1C and D*). Palm-Bassoon does not appear to localize in the ER, Golgi apparatus, or lysosomes (*Figure 1—figure supplement 1D, E, and F*). Comparing the ratio of Bassoon signal intensity at the periphery vs inside of the cell in randomly selected single sections from confocal stacks of palm-Bassoon and Bassoon-transfected cells (N=10 cells, 3 transfections per group, *Figure 1E*) demonstrated a higher Bassoon signal intensity at the periphery of cells expressing the palm-Bassoon construct (****p<0.0001, Mann-Whitney-Wilcoxon test), implying successful plasma membrane targeting of Bassoon.

Next, we co-expressed palm-Bassoon and RIBEYE in HEK293 cells and observed colocalizing RIBEYE and Bassoon immunofluorescent puncta at the periphery of cells (7 transfections; *Figure 2Ai*). Closer inspection of these immunofluorescent puncta with stimulated emission depletion (STED) nanoscopy (*Figure 2Aii*) revealed discrete structures typically consisting of ellipsoidal RIBEYE clusters juxtaposing on top of plate-like palm-Bassoon structures which seemingly anchor the RIBEYE clusters to the plasma membrane. We found the morphology of the RIBEYE+palm-Bassoon structures to be strikingly reminiscent of the arrangement of the two proteins in IHC ribbon synapses, where an ellipsoid/spherical synaptic ribbon composed of RIBEYE is found seated on a Bassoon plate that anchors it to the presynaptic AZ membrane (e.g. *Wong et al., 2014*; *Michanski et al., 2019*; *Michanski et al., 2023*; *Figure 2Bi and ii*; data from *Michanski et al., 2023*).

We compared the RIBEYE and Bassoon signal intensities at the periphery vs inside of the cell in randomly selected single sections from confocal stacks of HEK cells expressing RIBEYE and palm-Bassoon (N=9 cells) and HEK cells expressing RIBEYE and full-length Bassoon (N=9 cells). We found an increased peripheral distribution of both RIBEYE and Bassoon when using palm-Bassoon (*Figure 2C*, ****p<0.0001, Mann-Whitney-Wilcoxon test), implying successful membrane targeting of RIBEYE by palm-Bassoon. In each transfection, ~10% cells showed co-expression of both RIBEYE and palm-Bassoon (for representative sample overview, see *Figure 2—figure supplement 1*). Of those, cells expressing discrete RIBEYE-palm Bassoon structures were discernible by the characteristic RIBEYE distribution along the periphery of the cell. This peripheral distribution, in turn, seemingly depends upon RIBEYE and palm-Bassoon expression ratios (shown in *Figure 2—figure supplement 1B*) as cells with little to no palm-Bassoon expression show predominantly cytosolic RIBEYE puncta and were not used for analysis. For simplicity, we henceforth refer to the structures composed of RIBEYE and palm-Bassoon in HEK293 cells as *SyRibbons* (for *synthetic* ribbons).

We next performed 3D surface renderings using Imaris 9.6 (Oxford Instruments) to assess these structures. In a given cell, only structures with colocalizing RIBEYE and palm-Bassoon immunofluorescence

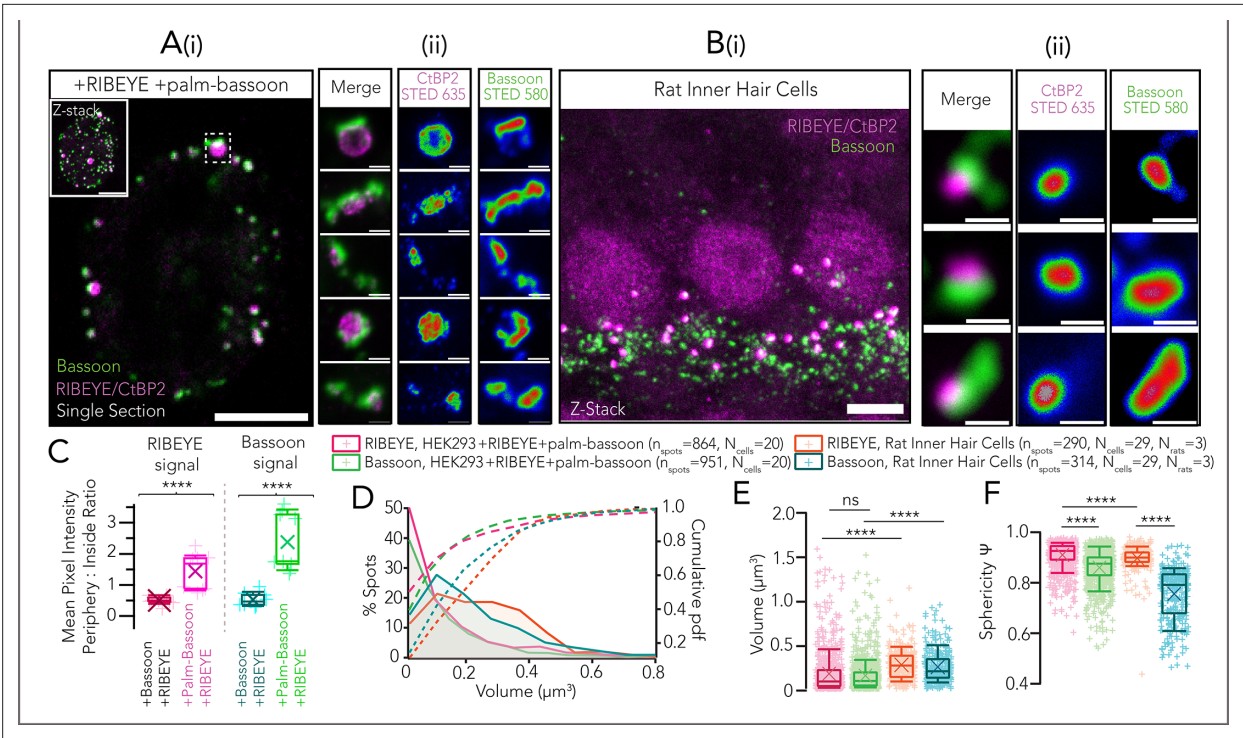

**Figure 2.** Co-expression of RIBEYE with palm-Bassoon results in ribbon-type active zone (AZ)-like structures. (**A**) (**i**) Representative confocal image (single section) of a human embryonic kidney 293 (HEK293) cell transfected with RIBEYE-GFP (magenta) and palm-Bassoon (green). Co-expression of RIBEYE with palm-Bassoon targets RIBEYE to the cell membrane. Inset shows maximum projection. Scale bar = 5 μm. (ii) Exemplary 2D stimulated emission depletion (STED) images for RIBEYE – palm-Bassoon juxtapositions acquired from cells as shown in (**i**). Scale bar = 500 nm; individual channels have been depicted with an intensity-coded look-up table with warmer colors indicating higher intensity. (**i**) Representative maximum projection of confocal sections of apical organ of Corti from a Wistar rat (postnatal day 18); data as published in *Michanski et al., 2023*, stainings were for CtBP2/RIBEYE (magenta; labeling synaptic ribbons and inner hair cell [IHC] nuclei) and Bassoon (green; spots juxtaposing with ribbons represent IHC AZs, spots not juxtaposing with ribbons represent efferent synapses formed by lateral olivocochlear neurons onto SGN boutons). Scale bar = 5 μm. (**B**) (**ii**) Juxtaposing RIBEYE and Bassoon spots imaged in 2D STED and confocal mode, respectively. Scale bar = 500 nm. Note the striking resemblance to reconstituted RIBEYE+palm-Bassoon structures in HEK293 cells as shown in (**Aii**). STED images are from 3 sample transfections, representative of 7 total transfections; individual channels have been depicted with an intensity-coded look-up table with warmer colors indicating higher intensity. (**C**) Quantification of RIBEYE and Bassoon signal intensity at periphery vs inside of cell shows a higher peripheral distribution of RIBEYE and Bassoon in HEK293+RIBEYE+palm-Bassoon cells (N=9 cells) as compared to HEK293+RIBEYE+Bassoon cells (N=9 cells); ****p<0.0001, Mann-Whitney-Wilcoxon test. Overlaid data points represent individual cells, crosses represent mean values, central band indicates the median, whiskers represent 90/10 percentiles, and boxes represent 75/25 percentiles. Distribution of volumes of RIBEYE and palm-Bassoon puncta from HEK293 cells expressing RIBEYE and palm-Bassoon (n = 20 cells, quantifications from 5 sample transfections). Volumes of synaptic ribbons from rat IHCs (n = 29 cells, 3 rats) have been plotted for comparison. (**E**) Box plot depicting data from (**D**). Volumes of RIBEYE and palm-Bassoon puncta are comparable to each other (p>0.99, Kruskal-Wallis test with post hoc Dunn's multiple comparison), but on average, they are much smaller and considerably more variable when compared to volumes of RIBEYE and Bassoon puncta from rat IHCs, respectively (****p<0.0001, Kruskal-Wallis test with post hoc Dunn's multiple comparison). Overlaid plus signs represent individual spots, crosses represent mean values, central band indicates the median, whiskers represent 90/10 percentiles, and boxes represent 75/25 percentiles. Quantification of RIBEYE and Bassoon signal intensity at periphery vs inside of cell shows a higher peripheral distribution of RIBEYE and Bassoon in HEK293+RIBEYE+palm-Bassoon cells (N=9 cells) as compared to HEK293+RIBEYE+Bassoon cells (N=9 cells); ****p<0.0001, Mann-Whitney-Wilcoxon test. Overlaid data points represent individual cells, crosses represent mean values, central band indicates the median, whiskers represent 90/10 percentiles, and boxes represent 75/25 percentiles. (**F**) RIBEYE puncta in HEK cells expressing RIBEYE and palm-Bassoon (n_spots = 864, N = 20 cells) appear more spherical than Bassoon puncta in the same cells (n_spots = 961, N = 20 cells; ****p<0.0001, Kruskal-Wallis test with post hoc Dunn's multiple-comparison test). Note the similar trend in IHC synaptic ribbons where RIBEYE puncta are more spherical than Bassoon puncta (n_spots = 290 for RIBEYE and n_spots = 314 for Bassoon, N = 29 cells, 3 rats; ****p<0.0001, Kruskal-Wallis test with post hoc Dunn's multiple-comparison test). Overlaid plus signs represent individual spots, crosses represent mean values, central band indicates the median, whiskers represent 90/10 percentiles, and boxes represent 75/25 percentiles.

The online version of this article includes the following figure supplement(s) for figure 2:

**Figure supplement 1.** Membrane localization of RIBEYE depends on expression levels of palm-Bassoon.

were considered for analysis to exclude occasional non-membrane localized spots (on average $51.75\pm40.84$ RIBEYE surfaces colocalizing with Bassoon surfaces per cell; N=20 cells). The volumes of RIBEYE and palm-Bassoon surfaces of *SyRibbons* were smaller on average and more variable (average volume ± standard deviation [SD]=$0.19 \pm 0.23$ $\mu m^3$ with a coefficient of variation [CV]=1.23 for RIBEYE; $n_{spots} = 864$ and volume = $0.17 \pm 0.18$ $\mu m^3$ with CV = 1.10 for Bassoon; $n_{spots} = 951$; data from N=20 cells, quantifications from 5 sample transfections) than volumes of synaptic ribbons and Bassoon immunofluorescent puncta from rat IHCs (volume = $0.29 \pm 0.17$ $\mu m^3$, CV = 0.59 for RIBEYE; $n_{spots} = 290$, and volume = $0.27 \pm 0.18$ $\mu m^3$, CV = 0.67 for Bassoon; $n_{spots} = 314$, data from N=29 cells) (*Figure 2D and E*). Moreover, volumes of RIBEYE surfaces show a high positive correlation to volumes of corresponding palm-Bassoon surfaces ($P_r = 0.778$, ****p<0.0001), implying that larger palm-Bassoon structures may recruit bigger RIBEYE structures to the plasma membrane. We also note that the volume of RIBEYE surfaces in *SyRibbons* appears smaller and well regulated in contrast to the predominantly large, cytosolic RIBEYE assemblies in cells co-expressing RIBEYE and full-length Bassoon (volume = $0.30 \pm 0.36$ $\mu m^3$, $n_{spots} = 460$, N=10 cells, ****p<0.001, Mann-Whitney-Wilcoxon test). In turn, Bassoon clusters seemed regulated by co-expressed RIBEYE: Bassoon surfaces at the plasma membrane were larger at *SyRibbons* than in the absence of RIBEYE in cells only expressing palm-Bassoon (volume = $0.14 \pm 0.21$ $\mu m^3$, $n_{spots} = 2217$, N=13 cells, ****p<0.001, Mann-Whitney-Wilcoxon test). RIBEYE clusters constituting *SyRibbons* appeared more spherical than the plate-like Bassoon structures in the same cells (sphericity $\Psi=0.91 \pm 0.05$, $n_{spots} = 864$ for RIBEYE vs $\Psi=0.86 \pm 0.07$, $n_{spots} = 951$ for Bassoon, data from N=20 cells; ****p<0.0001, Kruskal-Wallis test with post hoc Dunn's multiple-comparison test). This follows the same trend as in rat IHCs where RIBEYE spots indeed appear more spherical ($\Psi=0.90 \pm 0.05$, $n_{spots} = 290$) compared to Bassoon spots ($\Psi=0.76 \pm 0.10$, $n_{spots} = 314$); ****p<0.0001, Kruskal-Wallis test with post hoc Dunn's multiple-comparison test,

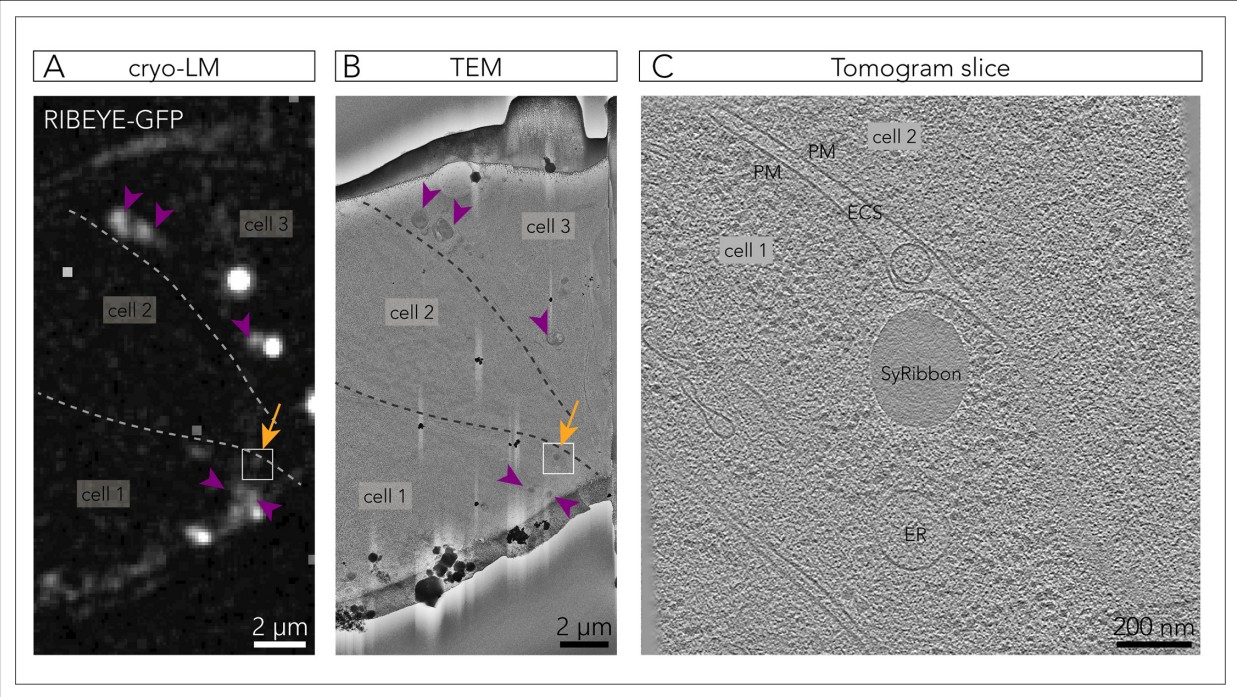

**Figure 3.** Cryo-correlative microscopy captures membrane-localized *SyRibbons*. (**A**) RIBEYE-GFP signal on a 150-nm-thick lamella, revealed by fluorescent light microscopy within the cryo-focused ion beam (cryo-FIB) chamber (cryo-LM). Dotted lines delineate cell membranes. The orange arrow indicates a plasma membrane-proximal GFP fluorescence, whereas purple arrowheads point to cytosolic RIBEYE aggregates. The tomogram shown in (**C**) was acquired at the boxed region. The image underwent background subtraction for better visualization. (**B**) Transmission electron microscopy (TEM) image of the lamella in (A). Orange arrow and purple arrowheads locate the respective regions in (**A**), showing that GFP-positive spots correlated to electron-dense bodies. Dotted lines delineate plasma membranes. The tomogram shown in (**C**) was taken at the boxed region. (**C**) Tomogram slice showing a *SyRibbon* acquired at the boxed region in (**B**). *PM*: plasma membrane, *ECS*: extracellular space, *ER*: endoplasmic reticulum.

The online version of this article includes the following figure supplement(s) for figure 3:

**Figure supplement 1.** Fluorescence-guided milling and tomogram acquisition of *SyRibbons*.

data from N=29 cells (*Figure 2F*). Nonetheless, the fact that next to structures with volumes comparable to IHC ribbons, we also encountered smaller and larger structures likely reflect poorer regulation of RIBEYE and palm-Bassoon expression in the heterologous system.

Next, we analyzed *SyRibbons* in situ using cryo-electron tomography (cryo-ET), which capitalizes on cell vitrification to obtain near-native preservation. Plunge-frozen HEK cells transfected with RIBEYE-GFP and untagged palm-Bassoon were vitrified and subsequently milled using a cryo-focused ion beam (cryo-FIB; *Figure 3—figure supplement 1A*) to produce 150-nm-thick lamellae as previously described (*Rigort et al., 2010*; *Pierson et al., 2024*, see Materials and methods). Using a light microscope integrated into the cryo-FIB chamber, we targeted cell areas showing peripheral GFP fluorescence corresponding to *SyRibbons* (*Figure 3—figure supplement 1B*). Lamellae were subsequently transferred to a cryo-transmission electron microscope, and tomographic tilt series were acquired at fluorescence locations. The tomograms showed that GFP-positive RIBEYE spots corresponded to electron-dense structures (*Figure 3A and B*), as is characteristic of the synaptic ribbon (*De Robertis and Franchi, 1956*; *Smith and Sjöstrand, 1961*). These electron-dense *SyRibbons* appeared to be ~300–800 nm in size and were ovoid or ellipsoidal in shape (*Figure 3B and C*, *Figure 3—figure supplement 1C*). Some *SyRibbons* displayed a hollow core (*Figure 3—figure supplement 1Ci and Ciii*) and on closer inspection, some appeared to have a multi-lamellar ultrastructure (*Figure 3—figure supplement 1Cii*), both of which have been reported previously for natively expressed synaptic ribbons in the inner ear and the retina (*Michanski et al., 2023*; *Michanski et al., 2019*; *Sobkowicz et al., 1982*; *Liberman, 1980*; *Stamataki et al., 2006*; *Wichmann and Moser, 2015*). We captured a *SyRibbon* positioned within 100 nm from the plasma membrane (*Figure 3C*). Although bona fide ribbons display a halo of SVs tethered on their surface, we did not observe any obvious accumulation of vesicular structures around these *SyRibbons*, which is not unexpected for synapse-naïve HEK cells.

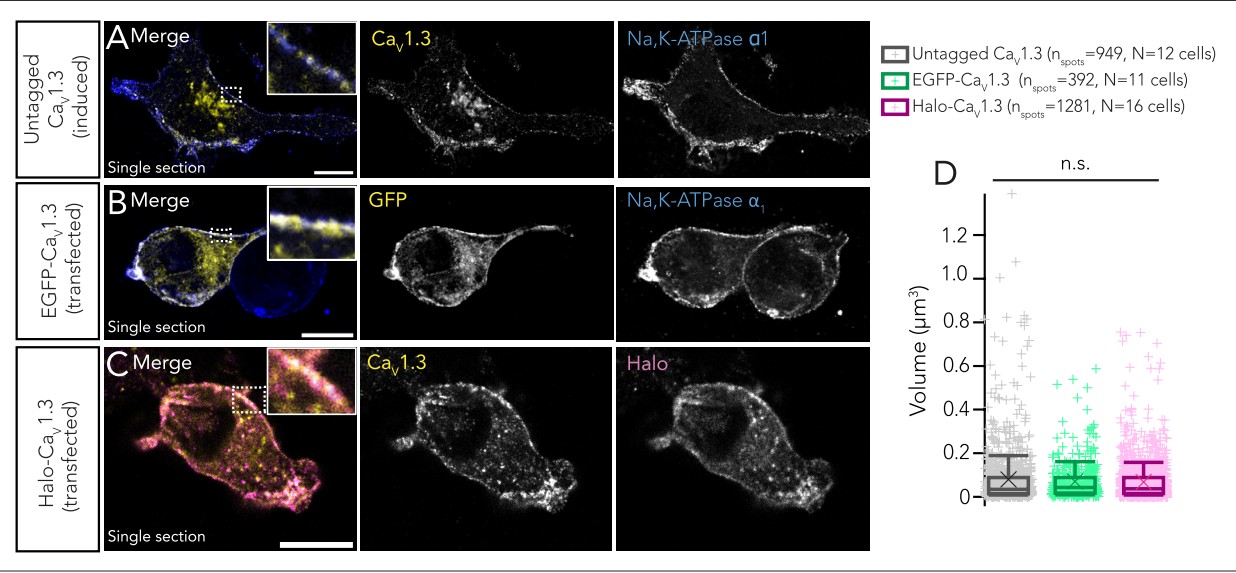

**Figure 4.** Human embryonic kidney 293 (HEK293) cells expressing $Ca_V1.3$. (**A**) Representative confocal section of a HEK293 cell stably expressing an inducible $Ca_V1.3\alpha$ transgene (untagged) along with constitutive transgenes for $Ca_V\beta_3$ and $Ca_V\alpha_2\delta_1$. Immunostainings have been performed using antibodies against $Ca_V1.3$ (green) and Na, K-ATPase $\alpha_1$ (blue, labeling the plasma membrane). Scale bar = 10 μm. (**B**) Representative confocal section of a HEK293 cell stably expressing $Ca_V\beta_3$ and $Ca_V\alpha_2\delta_1$, and transiently transfected with an N-terminal EGFP-tagged $Ca_V1.3$ construct. Immunostainings have been performed using antibodies against GFP (green) and Na, K-ATPase $\alpha_1$ (blue). Scale bar = 10 μm. (**C**) Representative confocal section of a HEK293 cell stably expressing $Ca_V\beta_3$ and $Ca_V\alpha_2\delta_1$ and transiently transfected with an N-terminal Halo-tagged $Ca_V1.3$ construct. Immunostainings have been performed using antibodies against $Ca_V1.3$ (green) and Halo-tag (magenta). Scale bar = 10 μm. Representative confocal section of a HEK293 cell stably expressing $Ca_V\beta_3$ and $Ca_V\alpha_2\delta_1$ and transiently transfected with an N-terminal EGFP-tagged $Ca_V1.3$ construct. Immunostainings have been performed using antibodies against GFP (green) and Na, K-ATPase $\alpha_1$ (blue). Scale bar = 10 μm. (**D**) Expression of either of the $Ca^{2+}$ channel complexes results in clusters of comparable sizes (p>0.05, Kruskal-Wallis test with post hoc Dunn's multiple-comparison test). Overlaid plus signs represent individual spots, crosses represent mean values, central band indicates the median, whiskers represent 90/10 percentiles, and boxes represent 75/25 percentiles.

The online version of this article includes the following figure supplement(s) for figure 4:

**Figure supplement 1.** $Ca_V1.3$ does not appear to colocalize directly with RIBEYE or with palm-Bassoon.

This is in contrast to a previous report in R28 retinal progenitor cells, where heterologously expressed RIBEYE was shown to recruit vesicles (*Magupalli et al., 2008*). Intriguingly, in some tomograms, we observed membrane-bound *SyRibbons* (*Figure 3—figure supplement 1Ci–iv*), which may indicate an autophagic engulfment of some of these overexpressed structures.

## RBP2 tethers Ca$_V$1.3 channels

To study the potential of *SyRibbons* to cluster Ca$^{2+}$ channels, we employed HEK293 cells stably expressing an inducible transgene of Ca$_V$1.3α1 along with constitutive transgenes for Ca$_V$β$_3$ and Ca$_V$α$_2$δ$_1$. We either used tetracycline for inducible Ca$_V$1.3α1 channel expression (*Figure 4A*) or performed transient transfections with Ca$_V$1.3 constructs containing an N-terminal EGFP- (*Figure 4B*) or Halo-tag (*Figure 4C*) for direct fluorescence imaging. Expression of either Ca$^{2+}$ channel complex resulted in small clusters at the plasma membrane as represented in *Figure 4D* (average volume 0.08±0.13 μm$^3$, n$_{spots}$ = 949, N=12 cells for untagged inducible Ca$_V$1.3α$_1$; 0.07±0.08 μm$^3$, n$_{spots}$ = 392, N=11 cells for EGFP-Ca$_V$1.3; and 0.07±0.09 μm$^3$, n$_{spots}$ = 1281, N=16 cells for Halo-Ca$_V$1.3α; P$_{untagged/EGFP}$=0.547, P$_{untagged/Halo}$>0.999, P$_{Halo/EGFP}$=0.582, Kruskal-Wallis test with post hoc Dunn's multiple-comparison test).

Co-expression of palm-Bassoon did not show colocalization with Ca$_V$1.3, which is in line with observations with full-length Bassoon (*Frank et al., 2010*) and the same applied when co-expressing RIBEYE and Ca$_V$1.3, arguing against a direct interaction for both (*Figure 4—figure supplement 1*). Rab-interacting molecule-binding protein (RIM-BP or RBP) links Bassoon to Ca$^{2+}$ channels (*Davydova et al., 2014*) and hence was an interesting candidate for tethering Ca$_V$1.3 to Bassoon clusters. RBP has been shown to interact with Ca$_V$1.3 (*Hibino et al., 2002*) and to be required for normal Ca$_V$1.3 Ca$^{2+}$ channel clustering at the IHC ribbon synapse (*Krinner et al., 2017*; *Krinner et al., 2021*). Indeed, when co-expressing RBP2 with palm-Bassoon and Ca$_V$1.3, we found juxtaposed palm-Bassoon and Ca$_V$1.3 both with immunolabeled (*Figure 5A*) and live-labeled (*Figure 5B and C*) proteins, indicating successful clustering of Ca$_V$1.3 at *synthetic* AZs.

We next performed ruptured whole-cell patch clamp to assess if RBP2 and/or palm-Bassoon or RIBEYE co-expression results in changes in Ca$_V$1.3 Ca$^{2+}$ currents. After tetracycline treatment for Ca$_V$1.3α$_1$ expression (18–24 hr), cells were transfected with both mKATE2-p2A-RBP2 and palm-Bassoon-GFP, with only palm-Bassoon-GFP, only mKATE2-p2A-RBP2, or RIBEYE-GFP constructs (*Figure 5—figure supplement 1A*). After 18–24 hr, the cell culture media was changed, and cells were recorded 24–36 hr afterward. We used 10 mM [Ca$^{2+}$]$_e$ and recorded Ca$^{2+}$ current (density)-voltage (IV) relations ~1 min after establishing the whole-cell configuration by applying step depolarizations of 20 ms from –86.2 to 58.8 mV in 5 mV increments. Ca$^{2+}$ current density appeared to be augmented upon expression of both RBP2 and palm-Bassoon when compared with induced-only controls (*Figure 5—figure supplement 1B and D*; \*\*p=0.0041, Kruskal-Wallis test with post hoc Dunn's multiple-comparison test). However, Ca$^{2+}$ current density in cells expressing only RBP2 or Palm-Bassoon did not significantly differ from either induced-only controls or cells expressing both proteins (p>0.05, Kruskal-Wallis test with post hoc Dunn's multiple-comparison test). For cells expressing only RIBEYE, we also did not observe any changes (p>0.999, Kruskal-Wallis test with post hoc Dunn's multiple-comparison test). We did not observe any noticeable differences in voltage dependence, activation kinetics, or inactivation kinetics of Ca$_V$1.3 Ca$^{2+}$ current upon expression of any of these AZ proteins (*Figure 5—figure supplement 1C, E, and F*; *Figure 5—figure supplement 2A and B*).

## Larger synthetic ribbon-type AZs establish larger Ca$_V$1.3 Ca$^{2+}$ channel clusters

Next, we tetra-transfected HEK293 cells with palm-Bassoon, Halo-tagged Ca$_V$1.3α$_1$, RBP2, and RIBEYE to explore the impact of co-expression of RIBEYE on the clustering of Ca$_V$1.3 Ca$^{2+}$ channels (*Figure 6A* for transfection scheme). Using confocal imaging of immunofluorescently labeled RIBEYE, Halo-Ca$_V$1.3, and Bassoon (RBP2 immunofluorescence was imaged using epifluorescence to select tetra-transfected HEK293 cells), we observed colocalization of Ca$_V$1.3α$_1$, Bassoon, and RIBEYE (*Figure 6B*). Line profiles drawn tangentially along the plasma membrane showed that beyond the general distribution of a Ca$_V$1.3α$_1$ signal in the plasma membrane, Ca$_V$1.3α$_1$ signal hotspots occurred underneath the *SyRibbons* (*Figure 6C*). The signal intensity of RIBEYE immunofluorescence positively correlated with the Ca$_V$1.3 signal immunofluorescence (P$_r$ = 0.839 and 0.886 from two sample line

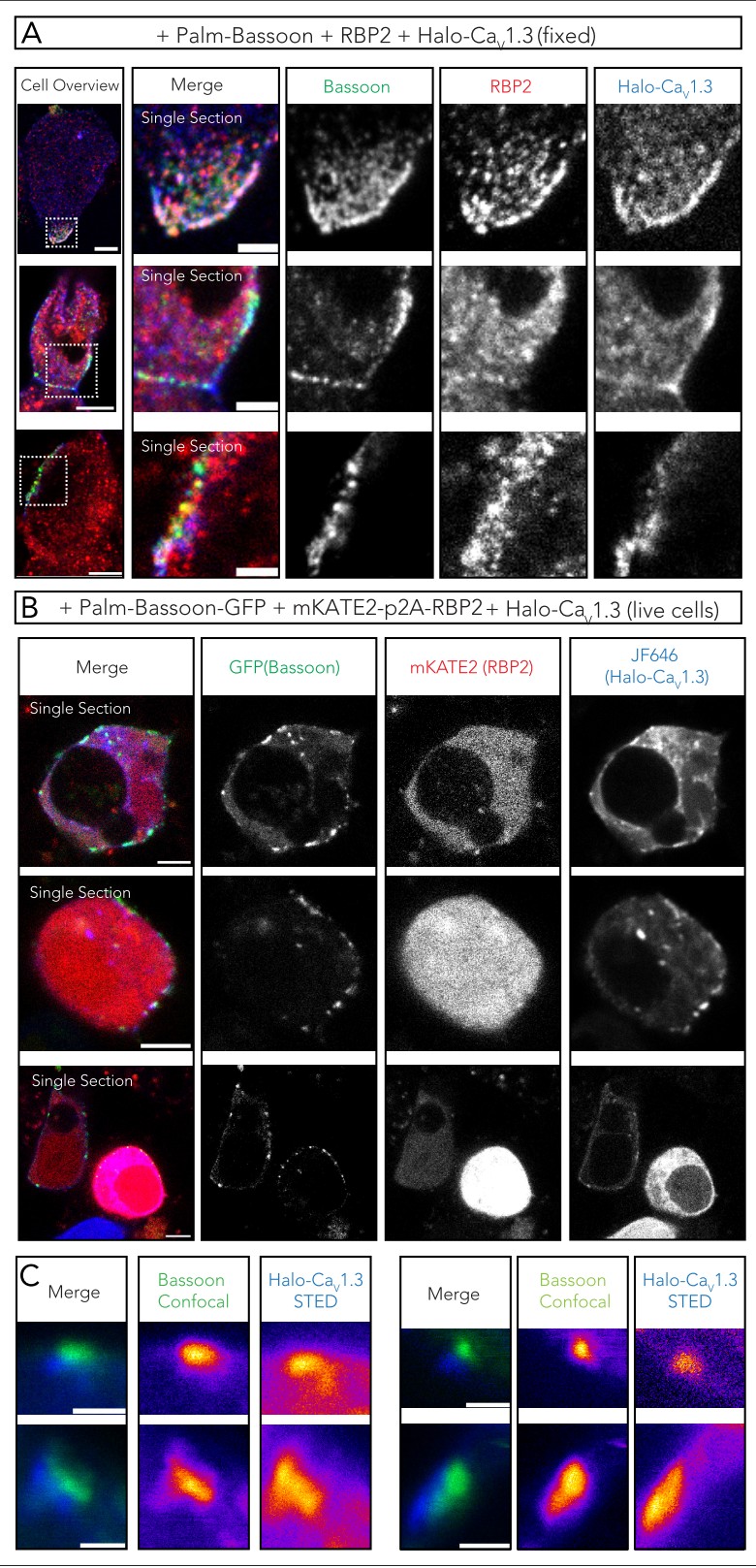

**Figure 5.** RBP2 bridges palm-Bassoon and Ca_V1.3 to form supramolecular assemblies at the plasma membrane. (**A**) Representative confocal images (single sections) of human embryonic kidney 293 (HEK293) cells (fixed) transfected with palm-Bassoon (green), RBP2 (red), and N-terminal Halo-tagged Ca_V1.3 (blue). The three colocalizing proteins appear to form supramolecular assemblies at the cell membrane. Scale bar = 5 µm for left

*Figure 5 continued on next page*

*Figure 5 continued*

panel showing cell overviews, 2 µm for right panels showing zoom-ins. (**B**) Representative confocal images (single sections) of live HEK293 cells transfected with palm-Bassoon-GFP (green), mKATE2-p2A-RBP2 (red, representing only mKATE2 signal and not RBP2 localization), and Halo-Ca$_V$1.3 (blue, labeled with Janelia Fluor 646 [JF646] HaloTag Ligand before imaging). Scale bar = 5 µm. (**C**) Exemplary super-resolution stimulated emission depletion (STED) images from live-labeled samples in (**B**) showing juxtaposition of palm-Bassoon and Halo-Ca$_V$1.3 signal at the plasma membrane in RBP2-positive cells. Scale bar = 1 µm. Representative confocal images (single sections) of HEK293 cells (fixed) transfected with palm-Bassoon (green), RBP2 (red), and N-terminal Halo-tagged Ca$_V$1.3 (blue). The three colocalizing proteins appear to form supramolecular assemblies at the cell membrane. Scale bar = 5 µm for left panel showing cell overviews, 2 µm for right panels showing zoom-ins.

The online version of this article includes the following figure supplement(s) for figure 5:

**Figure supplement 1.** Whole-cell Ca$^{2+}$ current amplitudes increase upon co-expression of RBP2 and palm-Bassoon.

**Figure supplement 2.** Kinetics of Ca$_V$1.3 channels are not altered upon co-expression of various active zone (AZ) proteins.

---

scans, \*\*\*\*p<0.0001), which has previously been reported for immunolabeled native ribbon-type AZs of mouse IHCs (***Ohn et al., 2016***). We further performed a pixel-based correlation analysis (***Figure 6—figure supplement 1A and B***), which indicated a moderate to strong colocalization of RIBEYE and Ca$_V$1.3 upon co-expression of palm-Bassoon and RBP2. In contrast, a poor colocalization was found in cells expressing only RIBEYE and Ca$_V$1.3; average Pearson's colocalization coefficient = 0.41 vs 0.24; average Mander's overlap coefficient = 0.63 vs 0.34, N=12 and 8 cells, respectively (\*\*p<0.01 m Wilcoxon's rank test). Colocalization quantifications for other transfection combinations shown in ***Figures 2 and 5***, ***Figure 4—figure supplement 1*** have been collectively depicted in ***Figure 6—figure supplement 1***. We next turned to two-color STED imaging, which provided improved spatial resolution and highlighted the confined localization of Ca$_V$1.3 Ca$^{2+}$ channel clusters underneath *SyRibbons*, as shown in ***Figure 6D***.

We performed surface renderings of confocally imaged Halo-tagged Ca$_V$1.3$\alpha_1$, RIBEYE, and palm-Bassoon immunofluorescence spots as described above for estimating the volume of *SyRibbons* and the Ca$_V$1.3 Ca$^{2+}$ channel clusters (***Figure 6E and F***). This revealed significantly larger Ca$_V$1.3 Ca$^{2+}$ channel clusters when colocalizing with *SyRibbons* in HEK293+Halo-Ca$_V$1.3+RBP2+*SyRibbons* cells (average volume ± SD = 0.18±0.28 µm$^3$, n$_{spots}$ = 634, N=13 cells, data representative of 5 transfections) compared to ribbonless (i.e. non-colocalized) Ca$_V$1.3 Ca$^{2+}$ channel clusters from the same cells (volume = 0.06 ± 0.10 µm$^3$, n$_{spots}$ = 1354, N=13 cells; \*\*\*\*p<0.0001, Kruskal-Wallis test with post hoc Dunn's multiple-comparison test) and Ca$_V$1.3 Ca$^{2+}$ clusters in HEK293 cells solely expressing Halo-Ca$_V$1.3 Ca$^{2+}$ channels (volume = 0.07 ± 0.09 µm$^3$, n$_{spots}$ = 1281, N=16 cells, data representative of 4 transfections; \*\*\*\*p<0.0001, Kruskal-Wallis test with post hoc Dunn's multiple-comparison test). Ribbonless Ca$_V$1.3 Ca$^{2+}$ channel clusters did not differ significantly regardless of co-expressing RIBEYE (p=0.099, Kruskal-Wallis test with post hoc Dunn's multiple-comparison test, see ***Figure 6F***). Moreover, volumes of Ca$_V$1.3 cluster surfaces show a moderate positive correlation with volumes of corresponding RIBEYE surfaces (P$_r$ = 0.385, \*\*\*\*p<0.0001), similar to previous observations at IHC ribbon synapses (***Frank et al., 2009***; ***Michanski et al., 2023***; ***Ohn et al., 2016***).

## Ca$^{2+}$ imaging reveals a partial spatial confinement of Ca$^{2+}$ signal at synthetic ribbon-type AZs

Finally, we set out to functionally characterize synthetic ribbon-type AZs. We combined patch-clamp recordings of Ca$_V$1.3 Ca$^{2+}$ influx with spinning-disk confocal microscopy to visualize Ca$^{2+}$ signals at the synthetic ribbon-type AZs in HEK293 cells. Following tetracycline induction for Ca$_V$1.3 expression, HEK293 cells were co-transfected with constructs expressing RIBEYE-GFP, untagged palm-Bassoon, and mKATE2-p2A-RBP2 (***Figure 7A***). We recorded mKATE2-positive cells that expressed peripheral GFP puncta, indicative of RIBEYE and palm-Bassoon co-expression. We employed peripheral GFP expression as a proxy of *SyRibbons* for functional analysis (***Figure 7B***). Ruptured patch-clamp recordings were performed with 10 mM [Ca$^{2+}$]$_e$ and 10 mM intracellular EGTA to enhance the signal-to-background ratio for visualizing Ca$^{2+}$ influx using the low-affinity red-shifted Ca$^{2+}$ indicator Calbryte590 (100 µM, k$_d$ = 1.4 µM). After loading the cell for approximately a minute, we first applied

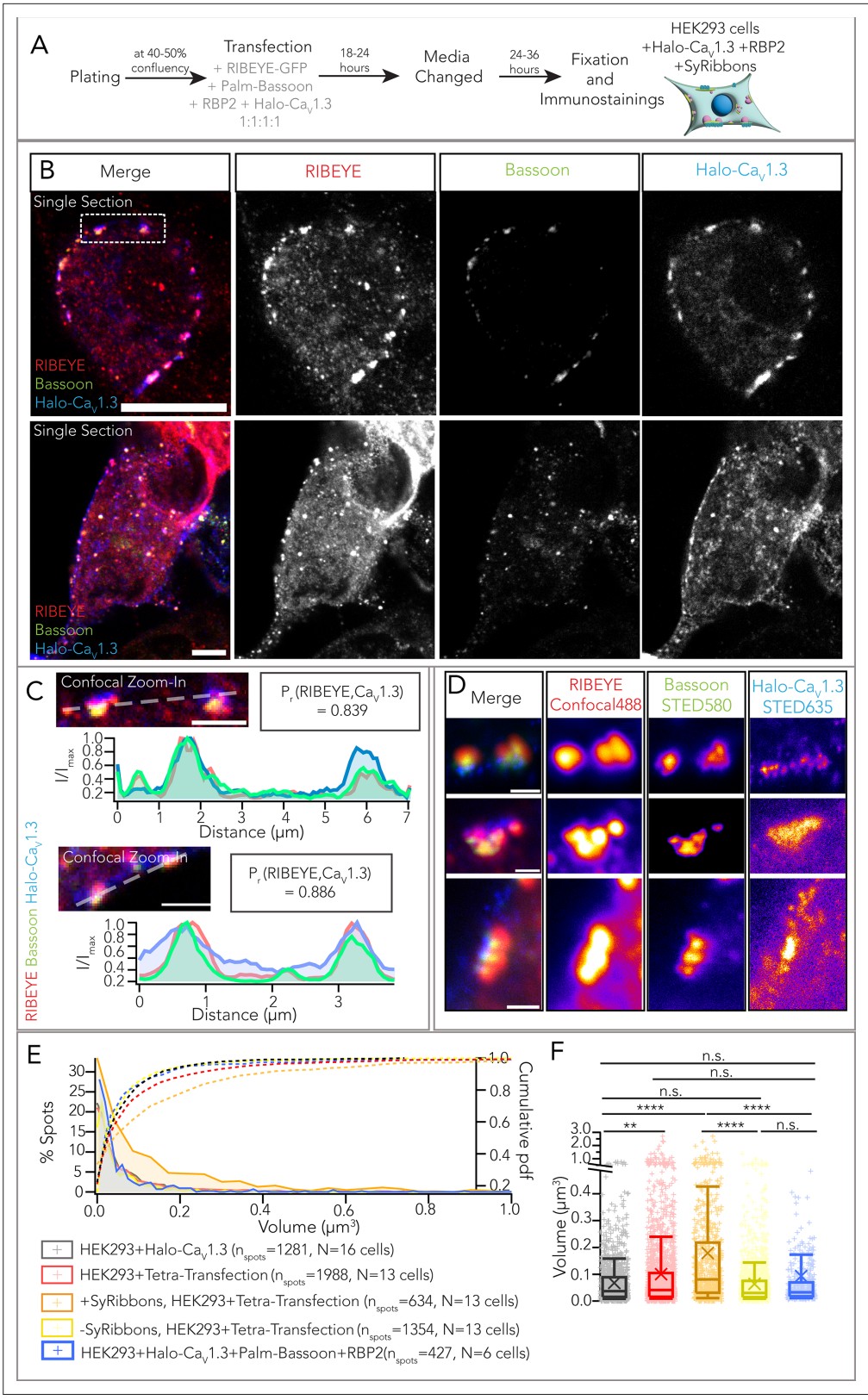

**Figure 6.** Synthetic ribbon-type active zones recruit $Ca_V1.3$ $Ca^{2+}$ channels. (**A**) Experimental scheme for expression of Halo-tagged $Ca_V1.3$, palm-Bassoon, RIBEYE-GFP, and RBP2 in human embryonic kidney 293 (HEK293) cells. (**B**) Representative confocal images (single sections) of HEK293 cells transfected with RBP2 (not shown, expression confirmed using epifluorescence), RIBEYE-GFP (red), palm-Bassoon (green), Halo-$Ca_V1.3$ (blue) shows

*Figure 6 continued on next page*

*Figure 6 continued*

colocalization of the three latter proteins at the plasma membrane. Scale bar = 5 µm. (**C**) Confocal zoom-ins from (**B**). Line scans depict an increased $Ca_V1.3$ signal intensity (blue) at sites where RIBEYE (red) and palm-Bassoon (green) clusters localize. Pearson's correlation coefficients ($P_r$) were calculated along the line profiles for RIBEYE and $Ca_V1.3$ signal intensity and indicate a high degree of correlation (~0.8) between the spatial localization of the two. Scale bar = 2 µm. (**D**) Super-resolution stimulated emission depletion (STED) images showing $Ca_V1.3$ clusters localizing at the base of the *SyRibbons*. Scale bar = 500 nm. (**E**) Distribution of volumes from confocal images of $Ca_V1.3$ clusters colocalizing with *SyRibbons* (orange, $n_{spots}$ = 634) and not colocalizing with *SyRibbons* (yellow, $n_{spots}$ = 1354) from HEK293 cells expressing $Ca_V1.3$, RBP2, and *SyRibbons* (N = 13 cells). Pooled volumes of all $Ca_V1.3$ clusters (with and without *SyRibbons*) from these cells have been shown in red ($n_{spots}$ = 1988). Volumes of $Ca_V1.3$ clusters from HEK293 cells expressing only Halo-$Ca_V1.3$ (black, $n_{spots}$ = 1281, N = 16 cells) and from cells expressing Halo-$Ca_V1.3$, RBP2, and palm-Bassoon (blue, $n_{spots}$ = 427, N = 6 cells) have been plotted for comparison. (**F**) Box plot depicting data from (**E**). $Ca_V1.3$ clusters appear larger on average in cells expressing $Ca_V1.3$, RBP2, and *SyRibbons* vs in cells expressing only $Ca_V1.3$ (**p=0.0044). Within cells expressing all four proteins, $Ca_V1.3$ clusters are much larger when colocalizing with *SyRibbons* vs when they do not (****p<0.0001). $Ca_V1.3$ clusters colocalizing with *SyRibbons* were in fact also larger than clusters in cells expressing only $Ca_V1.3$ and cells expressing only RBP2 and palm-Bassoon (****p<0.0001). Statistical test: Kruskal-Wallis test with post hoc Dunn's multiple-comparison correction. Overlaid plus signs represent individual spots, crosses represent mean values, central band indicates the median, whiskers represent 90/10 percentiles, and boxes represent 75/25 percentiles.

The online version of this article includes the following figure supplement(s) for figure 6:

**Figure supplement 1.** Colocalization quantifications.

step depolarizations of 20 ms from –83 to +62 mV in 5 mV step increments to measure $Ca^{2+}$ IV relations. We did not observe any changes in either the $Ca^{2+}$ current density (p=0.293, t-test) or the voltage dependence of $Ca^{2+}$ current (p=0.780, t-test when comparing $V_{half}$) when comparing cells expressing $Ca_V1.3$+RBP2+*SyRibbons* with cells expressing only $Ca_V1.3$ (***Figure 7C and D***). Next, we applied a depolarizing pulse to +2 mV for 500 ms and acquired images at a frame rate of 20 Hz (***Figure 7E***). The intracellular $Ca^{2+}$ signals mediated by $Ca_V1.3$ $Ca^{2+}$ influx were visualized as an increase in Calbryte590 fluorescence. The increase in Calbryte590 fluorescence appeared to spread across the HEK293 cell membrane, with discrete regions of higher signal intensity at *SyRibbons* (exemplary cell shown in ***Figure 7F*** with zoom-in at site with *SyRibbon* shown in ***Figure 7G***). We note that we did not find noticeable red fluorescence at *SyRibbons* in the absence of depolarization, which is similar to findings with Fluo-4FF in IHCs (***Frank et al., 2009***), but different from the situation for the large spherical presynaptic bodies in bullfrog hair cells that are stained by Fluo-3 (***Issa and Hudspeth, 1996***). Line profiles drawn tangentially to the membrane in composite ΔF image of Calbryte590 signal and RIBEYE-GFP showed a high correlation between the localization of *SyRibbons* and peak intensity of Calbryte590 fluorescence increase, indicating preferential $Ca^{2+}$ signaling at the *SyRibbons* (two representative line scans shown in ***Figure 7H***, Pearson's correlation coefficients of 0.783 and 0.757). Recordings were made from 24 cells (7 transfections) and regions of interest (ROIs) (diameter = 2 µm) with and without *SyRibbons* were analyzed as shown in ***Figure 7I***. Calbryte590 $ΔF_{max}/F_0$ values were calculated for each ROI which on average showed higher $ΔF_{max}/F_0$ values for ROIs with *SyRibbons* (on average ~26% higher) than for ROIs without *SyRibbons* (*p=0.018, paired t-test, ***Figure 7J***). We note that occasionally in some cells, we observed a higher Calbryte590 signal increase at sites without *SyRibbons*, and we attribute this to the likely presence of AZ-like clusters composed of RBP2 and palm-Bassoon in these regions, which we could not visualize.

## Discussion

In this study, we reconstituted and characterized a minimal ribbon-type AZ model system using heterologous expression. Co-expressing $Ca_V1.3$ $Ca^{2+}$ channels with membrane-targeted Bassoon for RIBEYE anchorage to the plasma membrane and attracting exogenous RBP2 for clustering $Ca^{2+}$ channels in HEK293 cells led to structures recapitulating basic aspects of IHC ribbon synapses. Despite the poor regulation of transgene expression in the synapse-naïve HEK293 cells, the AZ-like structures in the subset of cells were similar in morphology to the native IHC AZs. This applied to the *synthetic* ribbons, as well as to the AZ-like clusters of Bassoon, RBP2, and $Ca_V1.3$ $Ca^{2+}$ channels. However, all three components exhibited variability beyond the substantial natural heterogeneity found among

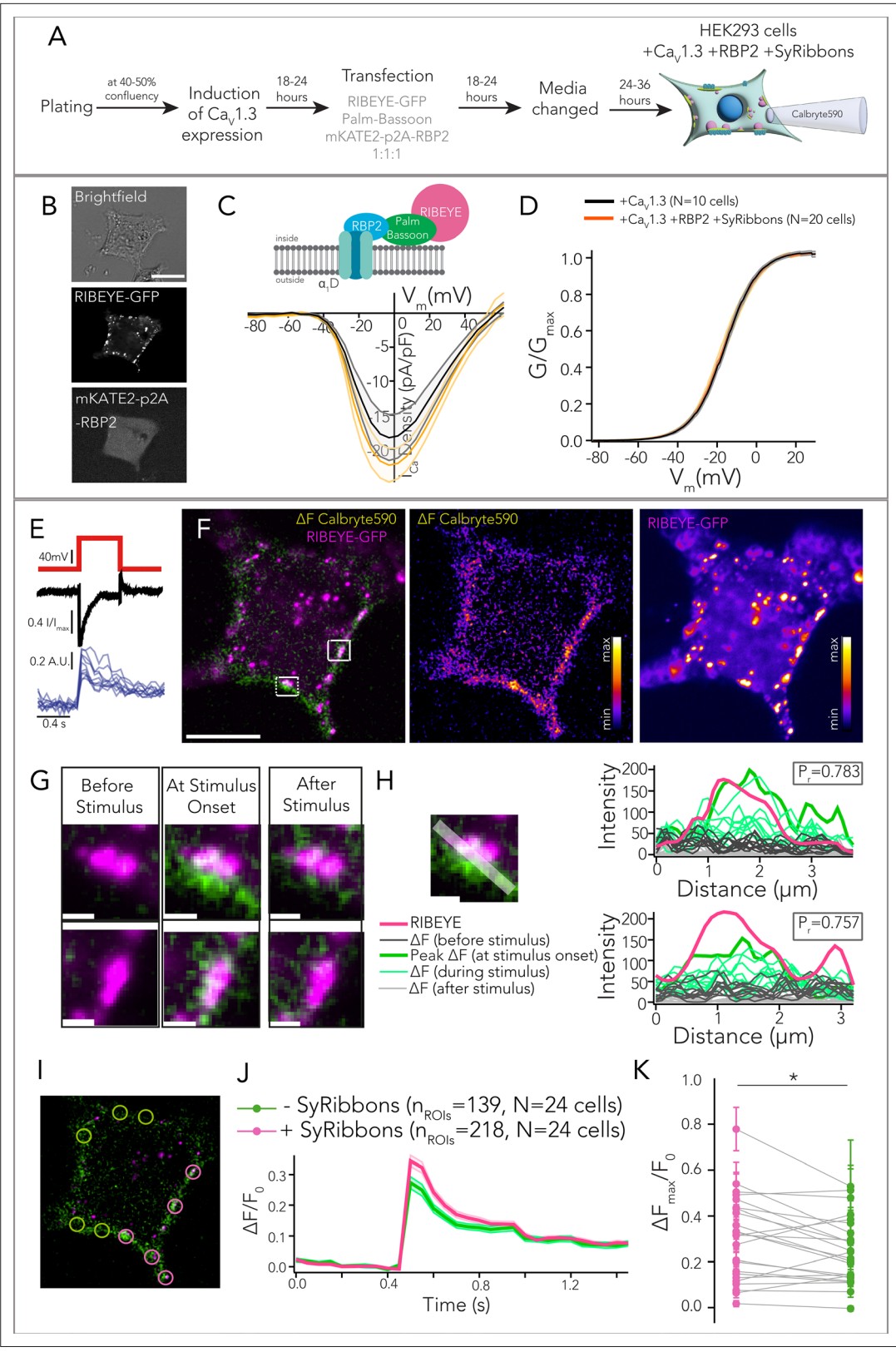

**Figure 7.** Ca²⁺ imaging reveals higher Ca²⁺ signal intensity underneath *SyRibbons*. (**A**) Experimental scheme for expression of *SyRibbons*, RBP2, and Ca$_V$1.3 in human embryonic kidney 293 (HEK293) cells for patch clamp in combination with Ca²⁺ imaging using the low-affinity Ca²⁺ indicator Calbryte590. (**B**) Exemplary cell used for Ca²⁺ imaging. Cells were identified by peripheral RIBEYE-GFP puncta as shown (indicative of RIBEYE and palm-Bassoon

*Figure 7 continued on next page*

*Figure 7 continued*

co-expression) and mKATE2 signal (indicative of RBP2 expression). (**C**) Current density-voltage (IV) relations from whole-cell patch clamp recordings of HEK293 cells expressing $Ca_V1.3$+RBP2+*SyRibbons* (n=20, in orange) and only $Ca_V1.3$ (n=10, in black). Lines represent mean current traces, and shaded area represents ± SEM. Whole-cell $Ca^{2+}$ current density does not appear to change upon co-expression of RBP2 and *SyRibbons* (p=0.293, t-test). (**D**) The voltage dependence of $Ca^{2+}$ current influx does not seem to be altered upon co-expression of RBP2 and *SyRibbons* at the whole-cell level. (**E**) A depolarizing pulse to +2 mV was applied to the cells for 500 ms, and the increase in Calbryte590 fluorescence was measured by acquiring images at a frame rate of 20 Hz using a spinning disk confocal microscope. (**F**) Representative image of a HEK293 cell expressing *SyRibbons*, RBP2, and $Ca_V1.3$ stimulated as described in (**E**). RIBEYE-GFP signal has been shown in magenta; the change in Calbryte590 fluorescence intensity (ΔF) upon $Ca^{2+}$ binding (the frame at the onset of depolarization to +2mV) has been shown in green. Note the distinct $Ca^{2+}$ signal 'hotspots' that colocalize with *SyRibbons*. Scale bar = 10 µm. (**G**) Zoom-ins from (F). Frames before, at the onset of, and after stimulation have been shown. Note the localized $Ca^{2+}$ influx at the base of the *SyRibbons*. Scale bar = 1 µm. (**H**) Line profiles drawn tangentially to the membrane in composite ΔF image of Calbryte590 signal and RIBEYE-GFP show a high correlation between the localization of *SyRibbons* and peak intensity of Calbryte590 fluorescence increase ($P_r$ = ~0.7). Plots show intensity profiles along line scans for RIBEYE-GFP (magenta) and Calbryte590 (light gray for frames before stimulus, bold green for peak intensity at onset of stimulus, light green lines for decaying intensity during ongoing stimulus, and dark gray for frames post-stimulus). (**I**) Regions of interest (ROIs) of 2 µm diameter were drawn at sites with (magenta circles) and without (green circles) *SyRibbons* and the corresponding Calbryte590 $ΔF/F_0$ values were calculated for each ROI. (**J**) Plot of average (shaded area represents ± SEM) $ΔF/F_0$ values from ROIs with and without *SyRibbons* (218 and 139 ROIs, respectively, from N = 24 cells, data from 7 transfections). The voltage dependence of $Ca^{2+}$ current influx does not seem to be altered upon co-expression of RBP2 and *SyRibbons* at the whole-cell level. (**K**) On average, Calbryte590 $ΔF_{max}/F_0$ was higher for ROIs with SyRibbons than ROIs without them in a given cell (*p=0.018, paired t-test). Dots represent mean of $ΔF_{max}/F_0$ from ROIs with (magenta) and without (green) SyRibbons averaged per cell, error bars represent ± SEM.

the IHC AZs (*Moser et al., 2023*). Although expression of RIBEYE does not seem to affect $Ca_V1.3$ physiology and kinetics, we demonstrate enhanced clustering of $Ca_V1.3$ channels and localization of $Ca^{2+}$ influx at *SyRibbons*, with a mild increase in $Ca^{2+}$ signal intensity. As *SyRibbons* partially resemble native hair cell AZs, we expect this easily available and experimentally accessible system to serve as an advanced testbed for functional interactions of AZ proteins and $Ca^{2+}$ channels. Further efforts toward reconstituting SVs and their release sites will help to enhance the utility of these synthetic AZs.

## Synthetic AZ reconstitution model to study ribbon synapse assembly

By demonstrating that membrane-targeted Bassoon suffices to anchor ribbon-like RIBEYE assemblies, this work adds to the top-down and bottom-up evidence for a key role of Bassoon in attracting RIBEYE to the presynaptic density at the AZ in photoreceptors and hair cells (*Dick et al., 2003*; *Khimich et al., 2005*; *Jing et al., 2013*; *tom Dieck et al., 2005*). Additional expression of multidomain proteins of the presynaptic density such as CAST (*Ohtsuka et al., 2002*; *Inoue et al., 2006*) might avoid the need for artificial palmitoylation of Bassoon for membrane targeting. Furthermore, our results support a model of a bidirectional control of AZ size and shape by RIBEYE and proteins of the presynaptic density. The amount of RIBEYE assembled in membrane-localized *SyRibbons* appeared regulated, strongly contrasting the seemingly unregulated cytosolic RIBEYE assemblies dominating HEK293 cells in the absence of membrane-targeted Bassoon. Vice versa, clusters of Bassoon at the plasma membrane were larger at *SyRibbons* than in the absence of RIBEYE. Finally, the size of *SyRibbons* seemingly scaled with the size of $Ca_V1.3$ and Bassoon clusters similar to what is observed in IHCs (*Ohn et al., 2016*). Such a model is consistent with results from genetic perturbation of native ribbon synapses: (i) disruption of RIBEYE leads to a disintegration of Bassoon into smaller subclusters in IHCs (*Jean et al., 2018*), and (ii) disruption of CAST and ELKS reduces the size of the ribbon-type AZs in rod photoreceptors (*tom Dieck et al., 2012*; *Hagiwara et al., 2018*). We speculate that achieving the full extent of large ribbon-type AZs such as in rod photoreceptors, as well as regulating the size of ribbon-type AZs at a specific synapse according to the precise functional demands, requires such a functional interplay between proteins of the presynaptic density and RIBEYE. We note that other ribbon-resident proteins such as Piccolino likely contribute to this fine-tuning of the size and shape of ribbon-type AZs, which will be an exciting question for future studies. Indeed, disruption of Piccolino in IHCs reduced the average ribbon size (*Müller et al., 2019*; *Michanski et al., 2023*). One exciting example of ribbon

synapse heterogeneity manifests itself in IHCs where synapses show a spatial size gradient that likely relates to the diverse molecular and functional properties of the postsynaptic SGNs (reviewed in *Moser et al., 2023*).

## Insights into presynaptic Ca²⁺ channel clustering and function

Current discussion of the clustering of Ca$_V$ Ca$^{2+}$ channels offers two different models: low-affinity protein-protein interactions (e.g. *Hibino et al., 2002*; *Kaeser et al., 2011*) via specific domains and liquid-liquid phase separation involving intrinsically disordered domains (e.g. *Heck et al., 2019*). Work on ribbon synapses has considered layers of organizing Ca$_V$ Ca$^{2+}$ channels at the AZ that seem more compatible with the former model: (i) 'micro-clustering' to which the synaptic ribbon contributes as a super-scaffold (*Jean et al., 2018*; *Neef et al., 2018*; *Frank et al., 2010*; *Maxeiner et al., 2016*) and (ii) tethering and 'nano-clustering' by RBPs, RIMs, CAST, and ELKS (*Liu et al., 2011*; *Grabner et al., 2015*; *Jung et al., 2015*; *Krinner et al., 2017*; *Luo et al., 2017*; *Hagiwara et al., 2018*). Moreover, there is converging evidence for nanoscale coupling of few Ca$_V$ Ca$^{2+}$ channels to SV release sites (*Brandt et al., 2005*; *Jarsky et al., 2010*; *Wong et al., 2014*; *Maxeiner et al., 2016*; *Özçete and Moser, 2021*; *Grabner and Moser, 2021*; *Jaime Tobón and Moser, 2023*) for which a role of the ribbon has been more controversial (e.g. *Maxeiner et al., 2016*; *Grabner and Moser, 2021*). MINFLUX nanoscopy of rod photoreceptor AZs recently revealed a well-ordered double-line array topography of Ca$_V$ Ca$^{2+}$ channels, RIM, Bassoon, and ubMunc13-2 at the presynaptic membrane on both sides of the ribbon (*Grabner et al., 2022*).

Clearly, we could not fully reconstitute this complex organization with the minimal set of molecular players at *SyRibbons* in HEK293 cells. Yet, we demonstrate a positive effect of *SyRibbons* on local Ca$^{2+}$ signaling, recapitulating micro-clustering of Ca$_V$1.3 Ca$^{2+}$ channels at IHC ribbon synapses. However, we note that the functional impact of ribbon-mediated clustering of Ca$_V$1.3 Ca$^{2+}$ channels remains unclear at the moment. Our Ca$^{2+}$ imaging data seems more reminiscent of previous work on ribbon-type AZs of immature IHCs (*Wong et al., 2014*), where a broader distribution of Ca$_V$1.3 and ribbons goes along with an immature state of the IHC AZs. We also note that the difference in Ca$^{2+}$ signaling near or apart from *SyRibbons* is rather subtle. We propose two explanations in this regard: (i) the use of an overexpression system diminishes the contrast of the Ca$^{2+}$ signal between AZ-like domains and the regular membrane. Overexpressed Ca$_V$1.3 is likely distributed diffusely across the plasma membrane in these cells along with more localized Ca$_V$1.3 clusters observed at *SyRibbons* (see *Figure 6B*, lower panel). (ii) While the presence of *SyRibbons* appears to promote the formation of larger Ca$_V$1.3 clusters and partially localize the Ca$^{2+}$ signal, it is unlikely that *SyRibbons* recruit a larger Ca$^{2+}$ channel complement. IHCs of RIBEYE KO mice were reported to show normal Ca$^{2+}$ current amplitudes at the whole-cell and single-AZ level (*Jean et al., 2018*), while the spatial spread of the Ca$^{2+}$ signal was more diffuse in the absence of the ribbon.

We propose that *SyRibbons* scaffold Ca$_V$1.3 channel nano-clusters in a non-native heterologous expression system, resulting in an increased channel density in regions with *SyRibbons*. In support of this hypothesis, the Ca$^{2+}$ signal averaged at *SyRibbons* showed a mildly increased maximal amplitude when compared with diffuse Ca$^{2+}$ channel distribution in the absence of *SyRibbons*. Instead, the total number of channels may be unaffected, which would explain why Ca$^{2+}$ current amplitudes at the whole-cell level are not impacted by expression of *SyRibbons*. A clear limitation in the study, therefore, arises from the use of an overexpression system, making it necessary to interpret results with caution and requiring in vivo and ex vivo validation. Future use of polycistronic gene expression or stable cell lines expressing all components could help with more consistent expression of the components and make the system become even more widely applicable.

Another limitation of the synthetic AZ system in HEK293 cells is the lack of SVs and the molecular machinery required for SV release. Nonetheless, our work paves the way for future studies, including stable co-expression of AZ proteins such as the release-site marker Munc13-1 (*Sakamoto et al., 2018*; *Böhme et al., 2016*) and potential delivery of SV machinery via co-expression (*Park et al., 2021*). Alternatively, using neurosecretory cells such as pheochromocytoma cells (PC12) or adrenal chromaffin cells in primary culture offers the advantage of providing synaptic-like microvesicles and a large set of components of the presynaptic machinery such that expression of exogenous proteins might be limited to RIBEYE. On the other hand, the complex molecular background involving various types of Ca$_V$ channels will likely complicate the interpretation. Regardless, *SyRibbons* in cultured cells offer

great availability and, as adherent cells, are well accessible to sophisticated analysis by techniques such as MINFLUX nanoscopy and cryo-ET.

# Materials and methods

## Key resources table

| Reagent type (species) or resource | Designation | Source or reference | Identifiers | Additional information |
|---|---|---|---|---|
| Cell Line (*Homo sapiens*) | ChanTest Stably Transfected Human Cav1.3/β3/α2δ-1-HEK293 cells | Charles River Laboratories, Cleveland, OH, USA | Cat. No. CT6232 | Tetracycline-inducible human $Ca_V1.3$ pore-forming $\alpha_1$-D subunit transgene (CACNA1D, NM_000720.2), constitutive expression of $Ca_V\beta_3$ (CACNB3, NM_000725.2) and $Ca_V\alpha_2\delta$–1 (CACNA2D1, NM_000722.2). Sex: HEK cells originate from a female fetus. |
| Recombinant DNA reagent | pAAV-CMV-HBA-RIBEYE-eGFP (plasmid) | This study | Addgene plasmid # 241988, RRID:Addgene_241988 | pAAV-GFP backbone, CMV-HBA promoter, human RIBEYE-GFP, Ampicillin resistance, WPRE regulatory element |
| Recombinant DNA reagent | pCS2+-CMV-Bassoon (plasmid) | This study | Addgene plasmid # 250160, RRID:Addgene 250160 | pCS2+ backbone, CMV promoter, rat Bassoon, Kanamycin resistance, SV40 poly(A) |
| Recombinant DNA reagent | C1-CMV-palm-Bassoon(95–3938)-EGFP (plasmid) | This study | Addgene plasmid # 250161, RRID:Addgene_250161 | pEGFP-C1 backbone, CMV enhancer, palm-rat Bassoon (95–3938)-EGFP, Kanamycin resistance, SV40 poly(A) |
| Recombinant DNA reagent | C1-CMV-palm-Bassoon(95–3938)-STOP (plasmid) | This study | Addgene plasmid # 250162, RRID:Addgene_250162 | pEGFP-C1 backbone, CMV enhancer, palm-rat Bassoon (95–3938)-STOP, Kanamycin resistance, SV40 poly(A) |
| Recombinant DNA reagent | pHTN-CMV-EGFP-$Ca_V$1.3 (plasmid) | *Schmitz et al., 2000* | – | pHTN backbone, CMV enhancer-promoter, EGFP-human *CACNA1D*, Ampicillin resistance, SV40 poly(A) |
| Recombinant DNA reagent | pHTN-CMV-HaloTag-$Ca_V$1.3 (plasmid) | *Schmitz et al., 2000* | – | pHTN backbone, CMV enhancer-promoter, HaloTag-human *CACNA1D*, Ampicillin resistance, SV40 poly(A) |
| Recombinant DNA reagent | f(syn)w-CMV-mKATE2-p2A-RBP2 (plasmid) | This study | Addgene plasmid # 241989, RRID:Addgene_241989 | F(syn)w backbone, CMV enhancer, mKATE2-p2A-RBP2, Ampicillin resistance, WPRE, SV40 poly(A) |
| Recombinant DNA reagent | pAAV-CMV-RBP2 (plasmid) | This study | Addgene plasmid # 241990, RRID:Addgene_241990 | pAAV backbone, CMV enhancer-HBA, RBP2, Ampicillin resistance, WPRE, SV40 poly(A) |
| Antibody | Anti-GFP (Chicken polyclonal) | Abcam | Ab13970, RRID:AB_300798 | IF(1:200) |
| Antibody | Anti-CtBP2 (Mouse monoclonal IgG1) | BD Biosciences | 612044, RRID:AB_399431 | IF(1:200) |
| Antibody | Anti-RIBEYE-A (Guinea pig polyclonal) | Synaptic Systems | 192104, RRID:AB_2800537 | IF(1:200) |
| Antibody | Anti-RIBEYE-A (Rabbit polyclonal) | Synaptic Systems | 192103, RRID:AB_2086775 | IF(1:200) |
| Antibody | Anti-RIBEYE-B (Rabbit polyclonal) | Synaptic Systems | 192003, RRID:AB_2261205 | IF(1:200) |
| Antibody | Anti-Bassoon (Chicken polyclonal) | Synaptic Systems | 141016 | IF(1:200) |
| Antibody | Anti-Bassoon (Mouse monoclonal IgG2a) | Abcam | Ab82958, RRID:AB_1860018 | IF(1:200) |
| Antibody | Anti-Na, K-ATPase $\alpha_1$ (Mouse monoclonal IgG1) | Abcam | Ab7671, RRID:AB_306023 | IF(1:200) |
| Antibody | Anti-Calnexin (Rabbit polyclonal) | Abcam | Ab22595, RRID:AB_2069006 | IF(1:200) |
| Antibody | Anti-GM130 (Mouse monoclonal IgG1) | BD Biosciences | 610822, RRID:AB_398142 | IF(1:200) |
| Antibody | Anti-LAMP1 (CD107A, Mouse monoclonal IgG1) | eBioscience | 14-1079-80, RRID:AB_467426 | IF(1:200) |
| Antibody | Anti-Halo (Mouse monoclonal) | Promega | G9211, RRID:AB_2688011 | IF(1:200) |

*Continued on next page*

*Continued*

| Reagent type (species) or resource | Designation | Source or reference | Identifiers | Additional information |
|---|---|---|---|---|
| Antibody | Anti-RBP2 (Rabbit polyclonal) | Synaptic Systems | 316103, RRID:AB_2619739 | IF(1:200) |
| Antibody | Anti-Ca$_V$1.3 (Rabbit polyclonal) | Alomone | ACC005, RRID:AB_2039775 | IF(1:100) |
| Antibody | Alexa Fluor 488 conjugated anti-chicken (Goat polyclonal) | Invitrogen | A11039, RRID:AB_2534096 | IF(1:200) |
| Antibody | Alexa Fluor 488 conjugated anti-guinea pig (Goat polyclonal) | Invitrogen | A11073, RRID:AB_2534117 | IF(1:200) |
| Antibody | Alexa Fluor 488 conjugated anti-rabbit (Goat polyclonal) | Invitrogen | A11008, RRID:AB_143165 | IF(1:200) |
| Antibody | Alexa Fluor 488 conjugated anti-mouse (Goat polyclonal) | Invitrogen | A11001, RRID:AB_2534069 | IF(1:200) |
| Antibody | Alexa Fluor 405 conjugated anti-rabbit (Goat polyclonal) | Invitrogen | A31556, RRID:AB_221605 | IF(1:200) |
| Antibody | Alexa Fluor 546 conjugated anti-mouse (Goat polyclonal) | MoBiTec | A11003, RRID:AB_2534071 | IF(1:200) |
| Antibody | Alexa Fluor 568 conjugated anti-chicken (Goat polyclonal) | Abcam | Ab175711, RRID:AB_2827757 | IF(1:200) |
| Antibody | Alexa Fluor 568 conjugated anti-guinea pig (Goat polyclonal) | Invitrogen | A11075, RRID:AB_2534119 | IF(1:200) |
| Antibody | Alexa Fluor 568 conjugated anti-rabbit (Goat polyclonal) | Thermo Fisher | A11011, RRID:AB_143157 | IF(1:200) |
| Antibody | Alexa Fluor 647 conjugated anti-mouse (Goat polyclonal) | Invitrogen | A21236, RRID:AB_2535805 | IF(1:200) |
| Antibody | Alexa Fluor 647 conjugated anti-rabbit (Goat polyclonal) | Invitrogen | A21244, RRID:AB_2535812 | IF(1:200) |
| Antibody | STAR Red conjugated anti-chicken (Goat monoclonal IgY) | Abberior | 2-0102-011-2 | IF(1:200) |
| Antibody | STAR 635p conjugated anti-mouse (Goat polyclonal) | Abberior | ST635P-1001-500UG, RRID:AB_2893232 | IF(1:200) |
| Antibody | STAR 635p conjugated anti-rabbit (Goat polyclonal) | Abberior | 2-0012-007-2 | IF(1:200) |
| Antibody | STAR 580 conjugated anti-mouse (Goat polyclonal) | Abberior | ST580-1001-500UG, RRID:AB_2923543 | IF(1:200) |
| Software, algorithm | ImageJ, Fiji | NIH | RRID:SCR_003070 | https://imagej.nih.gov/ij/, http://fiji.sc |
| Software, algorithm | Illustrator | Adobe | RRID:SCR_010279 | |
| Software, algorithm | Imaris 9.6 | Oxford Instruments | RRID:SCR_007370 | http://www.bitplane.com/imaris/imaris |
| Software, algorithm | Igor Pro 6 and 7 | WaveMetrics | RRID:SCR_000325 | http://www.wavemetrics.com/products/igorpro/igorpro.htm |
| Software, algorithm | Prism | GraphPad | RRID:SCR_015807 | https://www.graphpad.com/scientific-software/prism/ |
| Software, algorithm | PatchMaster | HEKA Electronics, Germany | RRID:SCR_000034 | http://www.heka.com/products/products_main.html#soft_pm |
| Software, algorithm | VisiView 5.0 | Visitron Systems GmbH | RRID:SCR_022546 | |
| Other | Janelia Fluor 646 HaloTag Ligand | Promega | Cat. No. GA1120 | Final concentration: 200 nM |
| Other | Calbryte590 | AAT Bioquest | Cat. No. 20706 | Effective concentration: 100 µM, $k_d$ = 1.4 µM |

## HEK cell culture and transfections

In this study, we used HEK293 cells stably expressing a tetracycline-inducible human $Ca_V1.3$ pore-forming $\alpha_1$-D subunit transgene (CACNA1D, NM_000720.2) and showing constitutive expression of $Ca_V\beta_3$ (CACNB3, NM_000725.2) and $Ca_V\alpha_2\delta$-1 (CACNA2D1, NM_000722.2). The stably expressing cell line was acquired from Charles River Laboratories, Cleveland, OH, USA (Cat. No. CT6232). The manufacturer's protocols report that pharmacological verification of the functional properties of the cloned channels was performed using IonWorks Barracuda (Molecular Devices). Absence of mycoplasma species in the line was verified using MycoAlert Kit (Lonza Rockland Inc). The cells were cultured as per product protocols in Dulbecco's modified Eagle medium (DMEM) containing GlutaMax, high glucose, and pyruvate (Gibco, Life Tech., 31966047), supplemented with 10% Fetal Bovine Serum (FBS; Gibco, Life Tech., A5256701) and 100 units/ml of Penicillin-Streptomycin (Gibco, Life Tech., 15070063). Additionally, the media was supplemented with 0.6 µM of Isradipine (Sigma-Aldrich, I6658) and the following selection antibiotics (in mg/ml): 0.005 Blasticidin (Invivogen, ant-bl-05), 0.50 Geneticin (G418 Sulfate; Gibco, Life Tech., 10131027), 0.10 Zeocin (InvivoGen, ant-zn-05), and 0.04 Hygromycin (Thermo Fisher, 10687010). Cells were cultured in a humidified incubator at 37°C with 5% (vol/vol) $CO_2$ saturation. Cells were split at ~70% confluency every 3–4 days to prevent adverse effects on cell growth and channel expression by dissociating the cell using Accutase (Sigma-Aldrich, A6964). The passage number did not exceed more than 26 passages. For experiments, cells were grown in media lacking selection antibiotics. For induction of $\alpha_1$-D subunit expression, cells were treated with selection antibiotic-free medium containing 3 µg/ml tetracycline. For transfection, 6.8 µg polyethylenimine (PEI, 25 kDa linear, Polysciences, 23966) was added along with a total of 2 µg of DNA to a final volume of 100 µl of DMEM (without FBS). The PEI/DNA mixture was thoroughly mixed and allowed to incubate for 30 min at room temperature before being added to the cells (30 µl/well in a 24-well plate containing 500 µl media, cell confluency ~50–70%). For all experiments with co-transfections, we used an equimolar ratio of DNA (total amount of DNA was kept 2 µg in a 100 µl transfection mix). After 18–24 hr, transfection media was replaced with fresh media devoid of selection antibiotics. Cells were used for experiments (immunocytochemistry, patch clamp, and $Ca^{2+}$ imaging) 24–48 hr after changing the media or as specified. If cell confluency was too high, cells were reseeded to an appropriate confluency and allowed to settle for at least 12 hr before commencement of experiments.

## Expression vectors for RIBEYE, Bassoon, $Ca_V1.3$, and RBP2

The RIBEYE-GFP construct used consisted of a human RIBEYE cDNA cloned in-frame into a pAAV-GFP vector driven by a hybrid CMV-enhancer-human-beta-actin promoter (CMV-HBA) and containing a downstream Woodchuck Hepatitis Virus (WHV) Posttranscriptional Regulatory Element (WPRE) for mRNA stabilization. All Bassoon constructs encode rat Bassoon. The untagged full-length constructs were cloned in the pCS2+ vector. For the generation of palm-Bassoon constructs, we used a pEGFP-C1 vector backbone with a CMV promoter and kanamycin resistance cassette. The insert comprises a palmitoylation consensus sequence of GAP43 (ATGCTGTGCTGTATGAGAAGAACC AAACAGGTTGAAAAGAATGATGAGGACCAAAAGATTTCCGGACTCAGATCTCGAG), followed by a cDNA sequence encoding amino acids 95–3938 of rat Bassoon and a C-terminal monomeric GFP. For the untagged version, the GFP tag was replaced by a STOP codon. To generate the Halo-$Ca_V1.3$ plasmid (*Schmitz et al., 2000*), human $Ca_V1.3$ cDNA (accession number NM_001128840.2) was de novo synthesized and assembled into a HaloTag vector (Promega G7721) using restriction cloning, thus encoding an N-terminal fusion of $Ca_V1.3$ to HaloTag linked by a 'GGS' sequence. The analogous GFP-$Ca_V1.3$ plasmid was generated by exchanging the HaloTag sequence for mEGFP using restriction cloning. The RBP2 construct comprises an insert encoding mouse RBP2 (accession ID NP_001074857.1) separated with a p2A cleavage site from an N-terminal mKATE2 tag. The mKATE2-p2A-RBP2 cassette was cloned in-frame into the f(syn)w-mKATE2-p2A vector, driven by a CMV promoter-enhancer and containing downstream WPRE sequence. The untagged version was made by cloning the RBP2 cDNA from this plasmid and replacing RIBEYE-GFP in the pAAV-CMV-HBA-RIBEYE-GFP-WPRE plasmid by in-fusion cloning (Takara Bio USA, Inc). Plasmids for expression of RIBEYE, Bassoon, and RBP2 have been deposited with AddGene and their RRIDs have been provided in the Key Resources Table.

## Immunocytochemistry and imaging

HEK293 cells plated on poly-L-lysine-coated coverslips were fixed with 99% chilled methanol at –20°C for 2 min as previously described (*Picher et al., 2017a*). The coverslips were washed thoroughly three times with PBS at room temperature (5–10 min). Blocking and permeabilization were performed with GSDB (goat serum dilution buffer: 16% normal goat serum, 450 mM NaCl, 0.3% Triton X-100, 20 mM phosphate buffer, pH ~7.4) for 45–60 min at room temperature. Samples were incubated with respective primary antibodies (diluted in GSDB, refer to Key Resources Table) overnight at 4°C or for 2 hr at room temperature. The samples were then washed three times (5–10 min each) with wash buffer (450 mM NaCl, 0.3% Triton X-100, 20 mM phosphate buffer, pH ~7.4). Incubation with appropriate secondary antibodies (also diluted in GSDB, refer to Key Resources Table) was performed for 1 hr at room temperature in a light-protected wet chamber. Lastly, the coverslips were washed three times with wash buffer (5–10 min each) and one final time with PBS, before mounting onto glass slides with a drop of fluorescence mounting medium (Mowiol 4-88, Carl Roth). For live-cell imaging of cells transfected with Halo-Ca$_V$1.3, cells plated on glass-bottom dishes were treated with Janelia Fluor 646 HaloTag Ligand (Promega, Cat. No. GA1120) at a final labeling concentration of 200 nM (in cell culture media). Cells were incubated with the ligand for ~60 min at 37°C with 5% (vol/vol) $CO_2$ saturation, after which the media was replaced with an equal volume of fresh, warm culture media.

Images from fixed and live samples were acquired in confocal/STED mode using an Olympus IX83 inverted microscope combined with an Abberior Instruments Expert Line STED microscope (Abberior Instruments GmbH). We used lasers at 488, 561, and 633 nm for excitation and a 775 nm (1.2 W) laser for STED. 1.4 NA 100× or 20× oil immersion objectives were used for fixed samples, and a water immersion 60× objective was used for live samples. Confocal stacks or single sections were acquired using Imspector software (pixel size = 80 × 80 nm along xy, 200 nm along z). For 2D-STED images, a pixel size of 15×15 nm (in xy) was used. Images in *Figure 2—figure supplement 1* were acquired using a Leica SP8 confocal microscope (Leica Microsystems, Germany). All acquired images were visualized using NIH ImageJ software and adjusted for brightness and contrast. Samples and their corresponding controls were processed in parallel using identical staining protocols, laser excitation powers, and microscope settings.

## Ultrastructural analysis using cryo-correlative light and microscopy

HEK293 cells were transfected with RIBEYE-GFP and untagged palm-Bassoon constructs as described above. The next day, the cells were detached using Accutase (Sigma-Aldrich, A6964) and replated on electron microscopy grids (R2/2, Au 200 mesh, 100 holey $SiO_2$ film; Quantifoil) at a density of 20,000 cells/grid. After 24 hr, grids were vitrified by plunge-freezing in liquid ethane/propane using a house-made manual plunger. Cryo-FIB milling and cryo-correlative light and electron microscopy were performed as previously described (*Rigort et al., 2010*; *Pierson et al., 2024*). In brief, upon transferring into an Aquilos 2 focused ion beam/scanning electron microscope (Thermo Fisher Scientific), the samples were imaged with an integrated fluorescent light microscope (Thermo Fisher Scientific) at 470 nm excitation wavelength to locate cells expressing peripheral GFP, which corresponded to membrane-localized *SyRibbons*. Milling was performed at a 9° angle at these areas, using ion currents between 500 and 100 nA. During milling, the GFP fluorescence was monitored additionally when the lamellae reached 2 µm and 150 nm thickness to ensure that the structures of interest were captured. All fluorescent data is shown as maximum intensity projections of 4 µm z-stacks. For visualization purposes, the fluorescent image shown in *Figure 3A* was corrected using the rolling ball background subtraction algorithm implemented in Fiji (*Schindelin et al., 2012*; *Sternberg, 1983*). After milling, lamellae were transferred into a Krios G4 cryo-transmission electron microscope (cryo-TEM; Thermo Fisher Scientific). Correlation of fluorescence on 150 nm lamellae and 8700× TEM overview revealed electron-dense structures at GFP puncta. Tilt series of these densities were collected at 33k magnification using a dose-symmetric scheme (from –45° to +63°, 9° pretilt, 3° increments) and with a 20 eV energy filter. The total electron dose was 120 e⁻/Å². The data was acquired using SerialEM (*Mastronarde, 2005*) running PACEtomo script (*Eisenstein et al., 2023*). Data pre-processing and tomogram reconstruction from tilt series were done automatically using an in-house script implementing MotionCor2, CTFFIND4, and IMOD (https://github.com/rubenlab/snartomo, copy archived at *Shaikh, 2025*; *Zheng et al., 2017*; *Rohou and Grigorieff, 2015*; *Mastronarde, 1997*). To enhance contrast, a

Wiener-like filter was applied (https://github.com/dtegunov/tom_deconv, copy archived at *Tegunov, 2019*) on the tomograms.

## Patch clamp and calcium imaging

$Ca_V1.3$ stably expressing HEK293 cells were plated on coverslips till about 40–50% confluency was achieved, following which they were induced for $Ca_V1.3\alpha_1$-D expression by incubating them for 24–48 hr in induction media supplemented with tetracycline. For transfected cells, transfection was performed after 24 hr of induction in the induction media as explained above. Cells were recorded in extracellular solution containing (in mM): 150 choline-Cl, 10 HEPES, 1 $MgCl_2$, and 10 $CaCl_2$; pH was adjusted to 7.3 with CsOH and osmolality was 300–310 mOsm/kg. For electrophysiological recordings in *Figure 5—figure supplements 1 and 2*, pipette solution contained (in mM): 140 NMDG, 10 NaCl, 10 HEPES, 5 EGTA, 3 Mg-ATP, and 1 $MgCl_2$. pH was adjusted to 7.3 with methanesulfonic acid and osmolality was around 290 mOsm/kg. For $Ca^{2+}$ imaging in *Figure 7*, the pipette solution contained (in mM): 111 Cs-glutamate, 1 $MgCl_2$, 1 $CaCl_2$, 10 EGTA, 13 TEA-Cl, 20 HEPES, 4 Mg-ATP, 0.3 Na-GTP, 1 L-glutathione, 0.1 Calbryte590 (AAT Bioquest, Cat. No. 20706); pH was adjusted to 7.3, osmolality to 290 mOsm/kg. Resistance of patch pipettes was 3–7 MΩ. HEK293 cells were rupture patch-clamped with EPC-10 amplifier (HEKA Electronics, Germany) controlled by Patchmaster software at room temperature, as described previously (*Picher et al., 2017b*). Cells were kept at a holding potential of –91.2 or –88 mV. All voltages were corrected for liquid junction potential offline (21.2 or 18 mV). Currents were leak-corrected using a p/10 protocol. Recordings were discarded when leak current exceeded –50 pA, $R_s$ exceeded 15 MΩ, or offset potential fluctuated more than 5 mV. IV relations were recorded around 1 min after rupturing the cell, by applying increasing step depolarization pulses of voltage ranging from –86.2 to 58.8 mV, in steps of 5 mV. Depolarization pulses of 500 ms were used for measuring $Ca_V1.3$ inactivation in *Figure 5—figure supplement 2*.

For $Ca^{2+}$ imaging experiments, we used a Yokogawa CSU-X1A spinning-disk confocal scanner, mounted on an upright microscope (Zeiss Examiner) with a 63×, 1.0 NA objective (W Plan-Apochromat, Zeiss). Images were acquired using an Andor Zyla sCMOS 4.2 camera, controlled by VisiView 5.0 software (Visitron Systems GmbH). GFP, mKATE2, and the low-affinity $Ca^{2+}$ indicator Calbryte590 were excited by a VS Laser Merge System (Visitron Systems) with 488 and 561 nm laser lines. Spinning disk was set to 5000 rpm. mKATE2-positive cells showing peripheral GFP expression (indicative of RIBEYE and palm-Bassoon co-expression) were identified and used for recordings. The cell was loaded for approximately a minute with Calbryte590 dye, after which we applied step depolarizations of 20 ms from –83 to +62 mV in 5 mV step increments to measure $Ca^{2+}$ IV relations. Next, a central plane of the cell was selected and the GFP fluorescence (green channel) of the cell showing *SyRibbons* was imaged. This was immediately followed by imaging of the increments of Calbryte590 fluorescence (red channel) in the same plane triggered by a depolarizing pulse to +2 mV for 500 ms (frame rate = 20 Hz). For $Ca^{2+}$ imaging experiments, we also included occasionally occurring HEK293 cells which were electrically coupled with other neighboring HEK293 cells. For such cells, we could not fully compensate for the slow capacitive currents, and we did not use them for electrophysiological analysis.

## Data analysis

### Analysis of confocal images

All images were processed using NIH ImageJ software to make z-projections and multi-channel composites, and figures were created using Adobe Illustrator. For analysis of membrane distribution, an ROI of 1 µm thickness was drawn along the periphery of the cell which was identified either using Na, K-ATPase $\alpha_1$ immunofluorescence or by enhancing the contrast in images of transfected cells so that the cell boundary becomes distinguishable. Mean pixel intensity was measured along this ROI and for the area enclosed within the ROI. The ratio of the mean pixel intensity (periphery: inside) per cell was reported as an average from at least three central planes; we did not use the basal and top plane. For volumetric fits of *SyRibbons*, IHC ribbons, and $Ca^{2+}$ channel clusters, we used the inbuilt surface detection algorithm from Imaris 9.6 (Oxford Instruments). The surface detail was set to 0.16 µm and largest sphere diameters for background subtraction were set as follows: for RIBEYE/CtBP2 0.3 µm, for Bassoon 0.1 µm, and for $Ca_V1.3$ 0.08 µm. Thresholding was performed based on the quality of immunofluorescence to ensure all discernible surfaces were detected. All surfaces with less than 10 voxels were filtered out. A manual check was done to include undetected surfaces, remove

surfaces not localizing within the cell of interest, and split surfaces that would be clubbed together. For analysis of *SyRibbons*, only RIBEYE surfaces colocalizing with palm-Bassoon surfaces were used for quantifications (at 0.06 µm or less). Occasionally, we would detect surfaces with volumes larger than 2 µm$^3$ (usually cytosolic) or smaller than 0.02 µm$^3$, which were not considered for analysis. For Ca$_V$1.3 cluster volumetric analysis in tetra-transfected cells, clusters classified as +*SyRibbons* were at 0.045 µm or less from palm-Bassoon-positive RIBEYE surfaces. All other clusters were classified as –*SyRibbons*. Only clusters localized at the periphery of the cell were used for analysis. Pixel-based intensity correlation analysis was performed using the *Coloc 2* plug-in on ImageJ (http://imagej.net/Coloc_2). Analysis was performed on 1-µm-thick ROIs marking the periphery of the cell in at least three central planes (excluding the basal and top plane). The Costes method was implemented with 100 repetitions to test for statistical significance of colocalization coefficients (*Costes et al., 2004*).

## Analysis of patch clamp and Ca$^{2+}$ imaging data

Analysis of electrophysiology data and figure preparation was performed using Igor Pro 6 and 7 (WaveMetrics Inc), and final figures were compiled using Adobe Illustrator. For analyzing current IV relations, the evoked Ca$^{2+}$ current was averaged from 5 to 10 ms after the onset of depolarization. Cells expressing current amplitudes less than –49 pA and with current density greater than –60 pA/pF were not taken into account for analysis.

Ca$^{2+}$ imaging data analysis was performed using a customized script in ImageJ. Briefly, we performed background subtraction in time series with Calbryte590 signal and then averaged 5 frames before stimulation and subtracted these from an average of three frames during stimulation to obtain ΔF images. For visualization, the images were smoothened (3×3 unweighted smoothening), and a composite was created by overlaying the ΔF images and the RIBEYE-GFP channel. For analyzing ΔF/F$_0$, we drew circular ROIs of 2 µm diameter at sites with and without *SyRibbons* in the background-subtracted time series. The ΔF$_{max}$/F$_0$ was calculated as the average of ΔF/F$_0$ in the first three frames of peak Ca$^{2+}$ signal intensity at stimulation onset in each ROI. To eliminate bias while assigning regions as with or without *SyRibbons*, ROIs were assigned in the RIBEYE-GFP channel, without looking at the corresponding Calbryte590 signal.

## Statistics

Data sorting and statistics were performed using MS Excel, Igor Pro 7, and/or GraphPad Prism 10. Numerical data is represented as mean ± standard deviation (SD). The normality of data was assessed using the Jarque-Bera test and the Kolmogorov-Smirnov test, and the equality of variances was checked using the *F*-test. When comparing two samples, a two-tailed unpaired Student's *t*-test (or paired *t*-test for *Figure 7J*) was performed for normally distributed samples with equal variances. If the conditions for normality and equality of variances were not met, an unpaired Mann-Whitney-Wilcoxon test was performed. When comparing multiple samples, a one-way ANOVA with post hoc Tukey's test was used for normally distributed data with equal variances. For samples that were not normally distributed, we used the Kruskal-Wallis test with post hoc Dunn's multiple-comparison correction. All box plots are depicted with crosses representing mean values, central bands indicating median, whiskers for 90/10 percentiles, boxes for 75/25 percentiles, and individual data points overlaid. Nonsignificant differences have been indicated as n.s., while significant p-values have been depicted as *p for <0.05, **p<0.01, ***p<0.001, and ****p<0.0001.

## Acknowledgements

We would like to thank Dr. Kathrin Kusch for advice on cloning, Dr. Vladan Rankovic and Dr. Maria Magdalena Picher for RIBEYE and RBP2 constructs, respectively, Dr. Jakob Neef for critical feedback on the manuscript, and Nare Karagulyan for valuable discussions regarding Ca$^{2+}$ imaging. We are grateful to Sandra Gerke, Christiane Senger-Freitag, Sina Langer, Fabienne Hahn, and Ina Preuss for expert technical assistance and Patricia Räke-Kügler for the administrative support during this study. We would also like to thank Tat Cheng for support with cryo-ET imaging and Prof. Erwin Neher and Prof. Silvio Rizzoli for their feedback throughout the course of this study. Lastly, we would like to thank lab rotation students Gantavya Arora and Ugur Coskun for their initial contributions to the project. RK was supported by funding from the Studienstiftung des Deutschen Volkes. This work was further supported by funding of the European Union (ERC, 'DynaHear', grant agreement No. 101054467)

and Fondation Pour l'Audition (FPA RD-2020-10) to TM and by Deutsche Forschungsgemeinschaft (DFG, German Research Foundation) via the EXC 2067/1 (MBExC) to TM, SEL, and RFB. Open access funding provided by Max Planck Society. Cryo-ET instrumentation was jointly funded by the DFG Major Research Instrumentation program (448415290) and the Ministry of Science and Culture of the State of Lower Saxony.

## Additional information

### Funding

| Funder | Grant reference number | Author |
|---|---|---|
| Deutsche Forschungsgemeinschaft | EXC 2067/1 (MBExC) | Stephan E Lehnart Rubén Fernández-Busnadiego Tobias Moser |
| Fondation Pour l'Audition | FPA RD-2020-10 | Tobias Moser |
| Studienstiftung des Deutschen Volkes | PhD Fellowship | Rohan Kapoor |
| Deutsche Forschungsgemeinschaft | DFG Major Research Instrumentation program 448415290 | Rubén Fernández-Busnadiego |
| Niedersächsisches Ministerium für Wissenschaft und Kultur | Cryo-ET instrumentation funding | Rubén Fernández-Busnadiego |
| European Research Council | 101054467 | Tobias Moser |

The funders had no role in study design, data collection and interpretation, or the decision to submit the work for publication. Open access funding provided by Max Planck Society.

### Author contributions

Rohan Kapoor, Conceptualization, Resources, Data curation, Software, Formal analysis, Funding acquisition, Validation, Investigation, Visualization, Methodology, Writing – original draft, Project administration, Writing – review and editing; Thanh Thao Do, Data curation, Formal analysis, Validation, Investigation, Visualization, Methodology, Writing – original draft, Writing – review and editing; Niko Schwenzer, Thomas Dresbach, Resources, Methodology, Writing – review and editing; Arsen Petrovic, Supervision, Validation, Methodology, Writing – review and editing; Stephan E Lehnart, Resources, Supervision, Methodology, Writing – review and editing; Rubén Fernández-Busnadiego, Supervision, Funding acquisition, Validation, Investigation, Visualization, Methodology, Writing – original draft, Writing – review and editing; Tobias Moser, Conceptualization, Resources, Formal analysis, Supervision, Funding acquisition, Validation, Investigation, Visualization, Methodology, Writing – original draft, Project administration, Writing – review and editing

### Author ORCIDs

Rohan Kapoor  https://orcid.org/0000-0002-4522-3289
Thanh Thao Do  https://orcid.org/0009-0004-2374-5803
Niko Schwenzer  https://orcid.org/0000-0003-0794-5501
Arsen Petrovic  https://orcid.org/0000-0003-4448-4230
Thomas Dresbach  https://orcid.org/0000-0002-2030-4787
Stephan E Lehnart  https://orcid.org/0000-0002-8115-3513
Rubén Fernández-Busnadiego  https://orcid.org/0000-0002-8366-7622
Tobias Moser  https://orcid.org/0000-0001-7145-0533

Reviewer #1 (Public review): https://doi.org/10.7554/eLife.98254.3.sa1
Reviewer #2 (Public review): https://doi.org/10.7554/eLife.98254.3.sa2

Reviewer #3 (Public review): https://doi.org/10.7554/eLife.98254.3.sa3

Author response https://doi.org/10.7554/eLife.98254.3.sa4

## Additional files

### Supplementary files
MDAR checklist

### Data availability
Source data for figures in this study are available at the Research Data Repository of the Göttingen Campus (GRO.data) with the https://doi.org/10.25625/3EH8MN.

The following dataset was generated:

| Author(s) | Year | Dataset title | Dataset URL | Database and Identifier |
|---|---|---|---|---|
| Kapoor R, Thao Do T, Schwenzer N, Petrovic A, Dresbach T, Lehnart SE, Fernández-Busnadiego R, Moser T | 2025 | Source Data Files for "Establishing synthetic ribbon-type active zones in a heterologous expression system | https://doi.org/10.25625/3EH8MN | Göttingen Research Online Data, 10.25625/3EH8MN |

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
