## [Editor Report · eLife Assessment]

The authors take a synthetic approach by introducing synaptic ribbon proteins into HEK cells to analyze how these assemblies cluster calcium channels at the active zone. Using a synapse-naive heterologous expression system and overexpression-based strategy is **valuable**, as it establishes a promising model for studying molecular interactions at the active zone. The study is built on a **solid** combination of super-resolution microscopy and electrophysiology, though it currently falls short of replicating the full functional properties of native ribbon synapses and instead resembles a multiprotein complex that partially mimics ribbon-type active zones.

---

## [Referee Report · Reviewer #1 (Public review)]

Summary:

In this manuscript, the authors attempt to reconstitute some active zone properties by introducing synaptic ribbon proteins into HEK cells. This "ground-up" approach can be valuable for assessing the necessity of specfic proteins in synaptic function. Here, the authors co-transfect a membrane-targeted bassoon, RBP2, calcium channel subunits and Ribeye to generate what they call "synthetic ribbons". The resultant structures show an ability to cluster calcium channels (Figure 4B) and a modest ability to concentrate calcium entry locations (figure 7J). At the light level, the ribeye aggregates look spherical and localize to the membrane through its interaction with the membrane-targeted bassoon and at the EM level the structures resemble those observed when Ribeye is overexpressed alone. It is a nice proof-of-principle in establishing a useful experimental system for studying calcium channel localization and with expression of other proteins perhaps a means to understanding structure and function of the ribbon. The paper does establish that previously described protein-interactions can be reconstituted in a heterologous system to and that the addition of Ribeye can increase the size of calcium channel patches via indirect interactions.

Strengths:

(1) The authors establish a new experimental system for the study of calcium channel localization to active zones.

(2) The clustering of calcium channels to bassoon via RBP2 is a nice confirmation of a previously-described interaction between bassoon and calcium channels in a cell-based system

(3) The "ground-up" approach is an attractive one and theoretically allows one learn a lot about the essential interactions for building a ribbon structure.

(4) The finding that introducing Ribeye can enhance the size of calcium channel patches is a novel finding that is interesting.

Weaknesses:

(1) The addition of EM is welcome, but the structures seem to resemble those created by overexpression of Ribeye alone, albeit at the membrane. It is unclear to me whether the interaction with Bsn or indirect interactions with other proteins has any effect on these structures. Also, while the abstract mentions that the size and shape are similar to ribbons, the EM seems to show that the size and shape are quite variable.

(2) The clustering of channels is accomplished by taking advantage of previously described interactions between RBP2, Ca channels and bassoon. While it is nice to see that it can be reconstituted in a naive cell, the interactions were previously described. The localization of Ribeye to bassoon takes advantage of a previously described interaction between the two and the membrane localization of the complexes required introduction of a membrane-anchoring motif. These factors limit the novelty of the findings.

(3) The difference in Ca imaging between SyRibbons and other locations is subtle. While there are reasonable explanations for why this could be the case, it may limit the utility of this system for studying Ca-channel-ribbon dynamics moving forward.

---

## [Referee Report · Reviewer #2 (Public review)]

Summary:

The authors show that co-expression of bassoon, RIBEYE, Cav1.3-alpha1, Cav-beta3, Cav-alpha2delta1, and RBP2 in a heterologus system (HEK293 cells) is sufficient to generate a protein complex resembling a presyanptic ribbon-type active zone both in morphology and in function (in clustering voltage-gated Ca channels and creating sites for localized Ca2+ entry). If the 3 separate Cav gene products are taken as a single protein (i.e. a Ca channel), the conclusion is that the core of a ribbon synapse comprises 4 proteins: bassoon holds the RIBEYE-containing ribbon to the plasma membrane, and RPB2 binds to bassoon and Ca channels, tethering the Ca channels to the presynaptic active zone.

Strengths:

(1) Good use of a heterologous system with generally appropriate controls provides convincing evidence that a presynaptic ribbon-type active zone (without the ability to support exocytosis), with the ability to support localized Ca2+ entry (a key feature of ribbon-type pre-synapses) can be assembled from a few proteins.

(2) In the revised manuscript, the authors do a good job of addressing the limitations of their cultured cell-system.

Weaknesses:

(1) Relies on over-expression, which almost certainly diminishes the experimentally-measured parameters (e.g. pre-synapse clustering, localization of Ca2+ entry).

(2) Are HEK cells the best model? HEK cells secrete substances and have a studied-endocytitic pathway, but they do not create neurosecretory vesicles. Initially, I asked why didn't the authors did not try to reconstitute a ribbon synapse in a cell that makes neurosecretory vesicles like a PC12 cell, and the authors addressed this question in their revision.

(3) Related to 1 and 2: the Ca channel localization observed is significant but not so striking given the presence of Cav protein and measurements of Ca2+ influx distributed across the membrane. Presumably, this is the result of overexpression and an absence of pathways for pre-synaptic targeting of Ca channels. But, still, it was surprising that Ca channel localization was so diffuse. I suppose that the authors tried to reduce the effect of over-expression by using an inducible Cav1.3? Even so, the accessory subunits were constitutively over-expressed.

---

## [Referee Report · Reviewer #3 (Public review)]

Summary:

Ribbon synapses are complex molecular assemblies responsible for synaptic vesicle trafficking in sensory cells of the eye and the inner ear. The Ca2+-dependent exocytosis occurs at the active zone (AZ), however, the molecular mechanisms orchestrating the structure and function of the AZs of ribbon synapses are not well understood. To advance in the understanding of those mechanisms, the authors present a novel and interesting experimental strategy pursuing the reconstitution of a minimal active zone of a ribbon synapse within a synapse-naïve cell line: HEK293 cells. The authors have used stably transfected HEK293 cells that express voltage-gated Ca2+ channels subunits (constitutive -CaV beta3 and CaV alpha2 beta1- and inducible CaV1.3 alpha1). They have expressed in those cells several proteins of the ribbon synapse active zone: (1) RIBEYE, (2) a modified version of Bassoon that binds to the plasma membrane through artificial palmitoylation (Palm-Bassoon) and (3) RIM-binding protein 2 (RBP2) to induce the formation of a minimal active zone that they called SyRibbons. The formation of such structures is convincing, however, the evidence of such structures having a functional impact (for example enhancing Ca2+-currents), as the authors claim, is weak. In conclusion, the novel approach shows that expression of a multiprotein complex partially reproduces properties, especially structural properties, of ribbon-type active zones in a heterologous system. Although the approach opens interesting possibilities for further experiments, the evidence supporting the functional properties of the so called "synthetic ribbon synapses" is incomplete.

Strengths of the study:

(1) The study is carefully carried out using a remarkable combination of (1) superresolution, correlative light microscopy and cryo-electron tomography, to analyze the formation and subcellular distribution of molecular assemblies and (2) functional assessment of voltage-gated Ca2+ channels using patch-clamp recording of Ca2+-currents and fluorometry to correlate Ca2+ influx with the molecular assemblies formed by AZ proteins. The results are of high quality and are in general accompanied of required control experiments.

(2) The method opens new opportunities to further investigate the minimal and basic properties of AZ proteins that are difficult to study using in vivo systems. The cells that operate through ribbon synapses (e.g. photoreceptors and hair cells) are particularly difficult to manipulate, so setting up and validating the use of a heterologous system more suitable for molecular manipulations is highly valuable.

(3) The structures formed by RIBEYE and Palm-Bassoon in HEK293 cells identified by STED nanoscopy and cryo-electron microscopy share relevant similarities similar to the AZs of ribbon synapses found in rat inner hair cells.

Weaknesses of the study:

(1) The evidence of the functional properties of the "synthetic ribbon-type active zones" has been only assessed by its effect on the modulation of Ca2+-channel function, and that effect is rather weak. The authors provide reasonable explanations regarding such a weak effect but, however, it is difficult to conclude that indeed the "synthetic ribbon-type active zones" are bona fide functional multiprotein complexes.

---

## [Author Response]

The following is the authors’ response to the original reviews

**Life Assessment**
The authors use a synthetic approach to introduce synaptic ribbon proteins into HEK cells and analyze the ability of the resulting assemblies to cluster calcium channels at the active zone. The use of this ground-up approach is valuable as it establishes a system to study molecular interactions at the active zone. The work relies on a solid combination of super-resolution microscopy and electrophysiology, but would benefit from: (i) additional ultrastructural analysis to establish ribbon formation in the absence of which the claim of these being synthetic ribbons might not be supported; (ii) data quantification to confirm colocalization of different proteins; (iii) stronger validation of impact on Ca2+ function; (iv) in depth discussion of problems derived from the use of an over-expression approach.

We thank the editors and the reviewers for the constructive comments and appreciation of our work. Please find a detailed point-to-point response below. In response to the critique received, we have now (i) included an ultrastructural analysis of the *SyRibbons* using correlative light microscopy and cryo-electron tomography, (ii) performed quantifications to confirm the colocalisation of the various proteins, (iii) discussed and carefully rephrased our interpretation of the role of the ribbon in modulating Ca^2+^ channel function and (iv) discussed concerns regarding the use of an overexpression system.

**Public Reviews:**

**Reviewer #1 (Public Review):**

We would like to thank the reviewer for the comments and advice to further improve our manuscript. We have completely overhauled the manuscript taking the suggestions of the reviewer into account.

(1) Are these truly "synthetic ribbons". The ribbon synapse is traditionally defined by its morphology at the EM level. To what extent these structures recapitulate ribbons is not shown. It has been previously shown that Ribeye forms aggregates on its own. Do these structures look any more ribbonlike than ribeye aggregates in the absence of its binding partners?

We thank reviewer 1 for their constructive feedback and critique of the work.

We agree that traditionally, ribbon synapses have always been defined by the distinct morphology observed at the EM level. However, since the discovery of the core-components of ribbons (RIBEYE and Piccolino) confocal and super-resolution imaging of immunofluorescently labelled ribbons have gained importance for analysing ribbon synapses. A correspondence of RIBEYE immunofluorescent structures at the active zone to electron microscopy observations of ribbons has been established in numerous studies (Wong et al, 2014; Michanski et al, 2019, 2023; Maxeiner et al, 2016; Jean et al, 2018) even though direct correlative approaches have yet to be performed to our knowledge. We have now analysed *SyRibbons* using cryo-correlative electron-light microscopy. We observe that GFPpositive RIBEYE spots corresponded well with electron-dense structures, as is characteristic for synaptic ribbons (Robertis & Franchi, 1956; Smith & Sjöstrand, 1961; Matthews & Fuchs, 2010). We could also observe *SyRibbons* within 100 nm of the plasma membrane (see Fig. 3). We have now added this qualitative ultrastructural analysis of *SyRibbons* in the main manuscript (lines 272 - 294, Fig. 3 and Supplementary Fig. 3).

(2) No new biology is discovered here. The clustering of channels is accomplished by taking advantage of previously described interactions between RBP2, Ca channels and bassoon. The localization of Ribeye to bassoon takes advantage of a previously described interaction between the two. Even the membrane localization of the complexes required the introduction of a membraneanchoring motif.

We respectfully disagree with the overall assessment. Our study emphasizes the synthetic establishment of protein assemblies that mimic key aspects of ribbon-type active zone, defining minimum molecular requirements. Numerous previous studies have described the role of the synaptic ribbon in organising the spatial arrangement of Ca^2+^ channels, regulating their abundance and possibly also modulating their physiological properties (Maxeiner et al, 2016; Frank et al, 2010; Jean et al, 2018; Wong et al, 2014; Grabner & Moser, 2021; Lv et al, 2016). We would like to highlight that there remain major gaps between existing in vitro and in vivo data; for instance, no evidence for direct or indirect interactions between Ca^2+^ channels and RIBEYE have been demonstrated so far. While we do indeed take advantage of previously known interactions between RIBEYE and Bassoon (tom Dieck et al, 2005); between Bassoon, RBP2 and P/Q-type Ca^2+^ channels (Davydova et al, 2014); and between RBP2 and Ltype Ca^2+^ channels (Hibino et al, 2002), our study tries to bridge these gaps by establishing the indirect link between the synaptic ribbon (RIBEYE) and L-type CaV1.3 Ca^2+^ channels using a bottom-up approach, which has previously just been speculative. Our data shows how even in a synapse-naive heterologous expression system, ribbon synapse components assemble Ca^2+^ channel clusters and even show a partial localisation of Ca^2+^ signal. Moreover, we argue that the established reconstitution approach provides other interesting insights such as laying ground-up evidence supporting the anchoring of the synaptic ribbon by Bassoon. Finally, we expect that the established system will serve future studies aimed at deciphering the role of putative CaV1.3 or CaV1.4 interacting proteins in regulating Ca^2+^ channels of ribbon synapses by providing a more realistic Ca^2+^ channel assembly that has been available in heterologous expression systems used so far. In response to the reviewers comment we have augmented the discussion accordingly.

(3) The only thing ribbon-specific about these "syn-ribbons" is the expression of ribeye and ribeye does not seem to participate in the localization of other proteins in these complexes. Bsn, Cav1.3 and RBP2 can be found in other neurons.

The synaptic ribbon made of RIBEYE is the key molecular difference in the molecular AZ ultrastructure of ribbon synapses in the eye and the ear. We hypothesize the ribbon to act as a superscaffold that enables AZ with large Ca^2+^ channel assemblies and readily releasable pools. In further support of this hypothesis, the present study on synthetic ribbons shows that CaV1.3 Ca^2+^ channel clusters are larger in the presence of *SyRibbons* compared to *SyRibbon*-less CaV1.3 Ca^2+^ channel clusters in tetratransfected HEK cells (Ca^2+^ channels, RBP, membrane-anchored Bassoon, and RIBEYE, Fig. 6). In response to the reviewers comment we now added an analysis of triple-transfected HEK cells (Ca^2+^ channels, RBP, membrane-anchored Bassoon), in which CaV1.3 Ca^2+^ channel clusters again are significantly smaller than at the *SyRibbons* and indistinguishable from *SyRibbon*-less CaV1.3 Ca^2+^ channel clusters (Fig. 6E, F).

(4) As the authors point out, RBP2 is not necessary for some Ca channel clustering in hair cells, yet seems to be essential for clustering to bassoon here.

Here we would like to clarify that RBP2 is indeed important in inner hair cells for promoting a larger complement of CaV1.3 and RBP2 KO mice show smaller CaV1.3 channel clusters and reduced whole cell and single-AZ Ca^2+^ influx amplitudes (Krinner et al, 2017). However, a key point of difference we emphasize on is that even though CaV1.3 clusters appeared smaller, they did not appear broken or fragmented as they do upon genetic perturbation of Bassoon (Frank et al, 2010), RIBEYE (Jean et al, 2018) or Piccolino (Michanski et al, 2023). This highlights how there may be a hierarchy in the spatial assembly of CaV1.3 channels at the inner hair cell ribbon synapse (also described in the discussion section “insights into presynaptic Ca^2+^ channel clustering and function”) with proteins like RBP2 regulating abundance of CaV1.3 channels at the synapse and organising them into smaller clusters – what we have termed as “nanoclustering”; while Bassoon and RIBEYE may serve as super-scaffolds further organizing these CaV1.3 nanoclusters into “microclusters”. Observations of fragmented Ca^2+^ channel clusters and broader spread of Ca^2+^ signal seen upon Ca^2+^ imaging in RIBEYE and Bassoon mutants (Jean et al, 2018; Frank et al, 2010; Neef et al, 2018), and the absence of such a phenotype in RBP2 mutants (Krinner et al, 2017) may be explained by such a differential role of these proteins in organising Ca^2+^ channel spatial assembly. The data of the present study on reconstituted ribbon containing AZs are in line with these observations in inner hair cells: RBP2 appears important to tether Ca^2+^ channels to Bassoon and these AZ-like assemblies are organised to their full extent by the presence of RIBEYE. As mentioned in the response to point 3 of the reviewer, we have now further strengthened this point by adding the analysis of *SyRibbon*-less CaV1.3 Ca^2+^ channel clusters in tripletransfected HEK cells (Ca^2+^ channels, RBP, membrane-anchored Bassoon, Fig. 6E, F). Moreover, we have revised the discussion accordingly.

(5) The difference in Ca imaging between SyRibbons and other locations is extremely subtle.

We agree with the reviewer on the modest increase in Ca^2+^ signal amplitude seen in the presence of *SyRibbons* and provide the following reasoning for this observation:

(i) It is plausible that due to the overexpression approach, Ca^2+^ channels (along with RBP2 and PalmBassoon) still show considerably high expression throughout the membrane even in regions where *SyRibbons* are not localised. Indeed, this is evident in the images shown in the lower panel in Fig. 6B, where Ca^2+^ channel immunofluorescence is distributed across the plasma membrane with larger clusters formed underneath *SyRibbons* (for an opposing scenario, please see the cell in Fig. 6B upper panel with very localised CaV1.3 distribution underneath *SyRibbons*). This would of course diminish the difference in the Ca^2+^ signals between membrane regions with and without *SyRibbons*. We note that while the contrast is greater for native synapses, extrasynaptic Ca^2+^ channels have been described in numerous studies alone for hair cells (Roberts et al, 1990; Brandt, 2005; Zampini et al, 2010; Wong et al, 2014).

(ii) Nevertheless, we do not expect a remarkably big difference in Ca^2+^ influx due to the presence of *SyRibbons* in the first place. Ribbon-less AZs in inner hair cells of RIBEYE KO mice showed normal Ca^2+^ current amplitudes at the whole-cell and the single-AZ level (Jean et al, 2018). However, it was the spatial spread of the Ca2+ signal at the single-AZ level which appeared to be broader and more diffuse in these mutants in the absence of the ribbon, in contrast to the more confined Ca2+ hotspots seen in the wild-type controls.

So, in agreement with these published observations – it appears that presence of *SyRibbons* helps in spatially confining the Ca^2+^ signal by super scaffolding nanoclusters into microclusters (see also our response to points 3 and 4 of the reviewer): this is evident from seeing some spatial confinement of Ca^2+^ signals near *SyRibbons* on top of the diffuse Ca^2+^ signal across the rest of the membrane as a result of overexpression in HEK cells.

We have now carefully rephrased our interpretation throughout the manuscript and added further explanation in the discussion section.

(6) The effect of the expression of palm-Bsn, RBP2 and the combination of the two on Ca-current is ambiguous. It appears that while the combination is larger than the control, it probably isn't significantly different from either of the other two alone (Fig 5). Moreover, expression of Ribeye + the other two showed no effect on Ca current (Figure 7). Also, why is the IV curve right shifted in Figure 7 vs Figure 5?

We agree with the reviewer that co-expression of palm-Bassoon and RBP2 seems to augment Ca^2+^ currents, while the additional expression of RIBEYE results in no change when compared to wild-type controls. We currently do not have an explanation for this observation and would refrain from making any claims without concrete evidence. As the reviewer also correctly pointed out, while the expression of the combination of palm-Bassoon and RBP2 raises Ca^2+^ currents, current amplitudes are not significantly different when compared to the individual expression of the two proteins (P > 0.05, Kruskal-Wallis test). In light of this, we have now carefully rephrased our MS. Moreover, we would like to thank reviewer 1 for pointing out the right shift in the IV curve which was due to an error in the values plotted on the x-axis. This has been corrected in the updated version of the manuscript.

(7) While some of the IHC is quantified, some of it is simply shown as single images. EV2, EV3 and Figure 4a in particular (4b looks convincing enough on its own, but could also benefit from a larger sample size and quantification)

We have now added quantifications for the colocalisations of the various transfection combinations depicted in the above-mentioned figures collectively in Supplementary Figure 7 and added the corresponding results and methods accordingly.

**Reviewer #2 (Public Review):**

We would like to thank the reviewer for the comments and advice to further improve our manuscript.

(1) Relies on over-expression, which almost certainly diminishes the experimentally-measured parameters (e.g. pre-synapse clustering, localization of Ca2+ entry).

We acknowledge this limitation highlighted by the reviewer arising from the use of an overexpression system and have carefully rephrased our interpretation and discussed possible caveats in the discussion section.

(2) Are HEK cells the best model? HEK cells secrete substances and have a studied-endocytitic pathway, but they do not create neurosecretory vesicles. Why didn't the authors try to reconstitute a ribbon synapse in a cell that makes neurosecretory vesicles like a PC12 cell?

This is a valid point for discussion that we also had here extensively. We indeed did consider pheochromocytoma cells (PC12 cells) for reconstitution of ribbon-type AZs and also performed initial experiments with these in the initial stages of the project. PC12 cells offer the advantage of providing synaptic-like microvesicles and also endogenously express several components of the presynaptic machinery such as Bassoon, RIM2, ELKS etc (Inoue et al, 2006) such that overexpression of exogenous AZ proteins would have to be limited to RIBEYE only.

However, a major drawback of PC12 cells as a model is the complex molecular background of these cells. We have also briefly described this in the discussion section (line 615 – 619). Naïve, undifferentiated PC12 cells show highly heterogeneous expression of various CaV channel types (Janigro et al, 1989); however, CaV1.3, the predominant type in ribbon synapses of the ear, does not seem to express in these cells (Liu et al, 1996). Furthermore, our attempts at performing immunostainings against CaV1.3 and at overexpressing CaV1.3 in PC12 cells did not prove successful and we decided on refraining from pursuing this further (data not shown).

On the contrary, HEK293 cells being “synapse-naïve” provide the advantage of serving as a “blank canvas” for performing such reconstitutions, e.g. they lack voltage-gated Ca^2+^ channels and multidomain proteins of the active zone. Moreover, an important practical aspect for our choice was the availability of the HEK293 cell line with stable (and inducible) expression of the CaV1.3 Ca^2+^ channel complex. Finally, as described in lines 613 – 614 of the discussion section, even though HEK293 cells lack SVs and the molecular machinery required for their release, our work paves way for future studies which could employ delivery of SV machinery via co-expression (Park et al, 2021) which could then be analyzed by the correlative light and electron microscopy workflow we worked out and added during revision.

(3) Related to 1 and 2: the Ca channel localization observed is significant but not so striking given the presence of Cav protein and measurements of Ca2+ influx distributed across the membrane. Presumably, this is the result of overexpression and an absence of pathways for pre-synaptic targeting of Ca channels. But, still, it was surprising that Ca channel localization was so diffuse. I suppose that the authors tried to reduce the effect of over-expression by using an inducible Cav1.3? Even so, the accessory subunits were constitutively over-expressed.

We agree with the reviewer on the modest increase in Ca^2+^ signal amplitude seen in the presence of *SyRibbons*. Yes, we employed inducible expression of the CaV1.3a subunit and tried to reduce the effect of overexpression by testing different induction times. However, we did not observe any major differences in expression and observed large variability in CaV1.3 expression across cells irrespective of induction duration. At all time points, there were cells with diffuse CaV1.3 localisation also in regions without *SyRibbons* which likely reduced the contrast of the Ca^2+^ signal we observe. We provide the following reasoning for this observation:

(i) It is plausible that due to the overexpression approach, Ca^2+^ channels (along with RBP2 and PalmBassoon) still show considerable expression along the membrane also in regions where *SyRibbons* are not localised. Indeed, this is evident in the images shown in the lower panel in Fig. 6B where Ca^2+^ channel immunofluorescence is distributed across the plasma membrane with larger clusters formed underneath *SyRibbons*. This would of course diminish the difference in the Ca^2+^ signals between membrane regions with and without *SyRibbons*. We note that while the contrast is greater for native synapses, extrasynaptic Ca^2+^ channels have been described in numerous studies alone for hair cells (Roberts et al, 1990; Brandt, 2005; Zampini et al, 2010; Wong et al, 2014).

(ii) Nevertheless, we do not expect a striking difference in Ca^2+^ influx amplitude due to the presence of *SyRibbons* in the first place. Ribbon-less AZs in inner hair cells of RIBEYE KO mice showed normal Ca^2+^ current amplitudes at the whole-cell and the single-AZ level (Jean et al, 2018). Instead, it was the spatial spread of the Ca^2+^ signal at the single-AZ level which appeared to be broader and more diffuse in these mutants in the absence of the ribbon, in contrast to the more confined Ca^2+^ hotspots seen in the wildtype controls.

So, in agreement with these published observations – it appears that presence of *SyRibbons* helps in spatially confining the Ca^2+^ signal by super scaffolding nanoclusters into microclusters: this is evident from seeing some spatial confinement of Ca^2+^ signals near SyRibbons on top of the diffuse Ca^2+^ signal across the rest of the membrane as a result of overexpression in HEK cells.

We have now carefully rephrased our interpretation throughout the manuscript and added further explanation in the discussion section.

**Reviewer #3 (Public Review):**

We would like to thank the reviewer for the comments and advice to further improve our manuscript.

(1) The results obtained in a heterologous system (HEK293 cells) need to be interpreted with caution. They will importantly speed the generation of models and hypothesis that will, however, require in vivo validation.

We acknowledge this limitation highlighted by Reviewer 3 arising from the use of an overexpression system and have carefully rephrased our interpretation and discussed possible caveats in the discussion section. We employed inducible expression of the CaV1.3a subunit and tried to reduce the effect of overexpression by testing different induction times. However, we did not observe any major differences in expression and observed large variability in CaV1.3 expression across cells irrespective of induction duration. At all time points, there were cells with diffuse CaV1.3 localisation, even in regions without SyRibbons and this could reduce the contrast of the Ca^2+^ signal we observe. We provide the following reasoning for this observation:

(i) It is plausible that due to the overexpression approach, Ca^2+^ channels (along with RBP2 and PalmBassoon) still show considerable expression along the membrane also in regions where *SyRibbons* are not localised. Indeed, this is evident in the images shown in the lower panel in Fig. 6B where Ca^2+^ channel immunofluorescence is distributed across the plasma membrane with larger clusters formed underneath *SyRibbons*. This would of course diminish the difference in the Ca^2+^ signals between membrane regions with and without *SyRibbons*. We note that while the contrast is greater for native synapses, extrasynaptic Ca^2+^ channels have been described in numerous studies alone for hair cells (Roberts et al, 1990; Brandt, 2005; Zampini et al, 2010; Wong et al, 2014).

(ii) Nevertheless, we do not expect a striking difference in Ca^2+^ influx amplitude due to the presence of *SyRibbons* in the first place. Ribbon-less AZs in inner hair cells of RIBEYE KO mice showed normal Ca^2+^ current amplitudes at the whole-cell and the single-AZ level (Jean et al, 2018). Instead, it was the spatial spread of the Ca^2+^ signal at the single-AZ level which appeared to be broader and more diffuse in these mutants in the absence of the ribbon, in contrast to the more confined Ca^2+^ hotspots seen in the wildtype controls.

So, in agreement with these published observations – it appears that presence of *SyRibbons* helps in spatially confining the Ca^2+^ signal by super scaffolding nanoclusters into microclusters: this is evident from seeing some spatial confinement of Ca^2+^ signals near SyRibbons on top of the diffuse Ca^2+^ signal across the rest of the membrane as a result of overexpression in HEK cells.

(2) The authors analyzed the distribution of RIBEYE clusters in different membrane compartments and correctly conclude that RIBEYE clusters are not trapped in any of those compartments, but it is soluble instead. The authors, however, did not carry out a similar analysis for Palm-Bassoon. It is therefore unknown if Palm-Bassoon binds to other membrane compartments besides the plasma membrane. That could occur because in non-neuronal cells GAP43 has been described to be in internal membrane compartments. This should be investigated to document the existence of ectopic internal Synribbons beyond the plasma membrane because it might have implications for interpreting functional data in case Ca2+-channels become part of those internal Synribbons.

In response to this valid concern, we have now included the suggested experiment in Supplementary Figure 1. We investigated the subcellular localisation of Palm-Bassoon and did not find Palm-Bassoon puncta to colocalise with ER, Golgi, or lysosomal markers, suggesting against a possible binding with membrane compartments inside the cell. We have added the following sentence in the results section, line 145 : “Palm-Bassoon does not appear to localize in the ER, Golgi apparatus or lysosomes (Supplementary Fig 1 D, E and F).”

(3) The co-expression of RBP2 and Palm-Bassoon induces a rather minor but significant increase in Ca2+-currents (Figure 5). Such an increase does not occur upon expression of (1) Palm-Bassoon alone, (2) RBP2 alone or (3) RIBEYE alone (Figure 5). Intriguingly, the concomitant expression of PalmBassoon, RBP2 and RIBEYE does not translate into an increase of Ca2+-currents either (Figure 7).

We agree with the reviewer that co-expression of palm-Bassoon and RBP2 seems to augment Ca^2+^ currents, while the additional expression of RIBEYE results in no change when compared to wild-type controls. We currently do not have an explanation for this observation and would refrain from making any claims without concrete evidence. We also highlight that, while the expression of the combination of palm-Bassoon and RBP2 raises Ca^2+^ currents, current amplitudes are not significantly different when compared to the individual expression of the two proteins (P > 0.05, Kruskal-Wallis test). In light of this, we have now carefully rephrased our MS.

(4) The authors claim that Ca2+-imaging reveals increased CA2+-signal intensity at synthetic ribbontype AZs. That claim is a subject of concern because the increase is rather small and it does not correlate with an increase in Ca2+-currents.

Thanks for the comment: please see our response to your first comment and the lines 585 – 610 in the discussion section.

**Recommendations for the authors:**

**Reviewer #2 (Recommendations For The Authors):**
(1) The authors should have a better discussion of problems derived from over-expression.

Done. Please see above.

(2) Ideally, the authors would repeat the study using a secretory cell line, but this is of course not possible. The idea could be brought forth, though.

As described above in our response to the public review of reviewer 2, we have discussed this idea in the discussion section (refer to lines 615 – 619), emphasizing on both the advantages and the limitations of using a secretory cell line (e.g. PC12 cells) instead of HEK293 cells as a model for performing such reconstitutions.

**Reviewer #3 (Recommendations For The Authors):**
(1) There are several figures in which colocalization between different proteins is studied only displaying images but without any quantitative data. This should be corrected by providing such a quantitative analysis.

We have now added quantifications for the colocalisations of the various transfection combinations depicted in the above-mentioned figures collectively in Supplementary Figure 7 and added the corresponding results and methods accordingly.

(2) The little increase in Ca2+-currents and Ca2+-influx associated to the clustering of Ca2+-channels to Synribbons is a concern. The authors should discuss if such a minor increase (found only when Palm-Bassoon and RBP2 ae co-expressed) would have or not physiological consequences in an actual synapse. They might discuss the comparison of those results and compare with results obtained in genetically modified mice in which Ca2+-currents are affected upon the removal of AZs proteins. On the other hand, they should explain why Ca2+-currents do not increase when the Synribbons are formed by RIBEYE, Palm-Bassoon and RBP2.

Done. Please see above.

(3) The description of the patch-clamp experiments should be enriched by including representative currents. Did the authors measure tail currents?

We would like to thank the reviewer for the valuable suggestion and have now added representative currents to the figures (see Supplementary Figure 5B). We agree with the reviewer on the importance of further characterizing the Ca^2+^ currents in the presence and absence of *SyRibbons* by analysis of tail currents for counting the number of Ca^2+^ channels by non-stationary fluctuation analysis but consider this to be out of scope of the current study and an objective for future studies.

(4) The current displayed in Figure 7 E should be explained better.

Previous studies have shown that Ca^2+^-binding proteins (CaBPs) compete with Calmodulin to reduce Ca^2+^-dependent inactivation (CDI) and promote sustained Ca^2+^ influx in Inner Hair Cells (Cui et al, 2007; Picher et al, 2017). In the absence of CaBPs, CaV1.3-mediated Ca^2+^ currents show more rapid CDI as in the case here upon heterologous expression in HEK cells ((Koschak et al, 2001), see also Picher et al 2017 where co-expression of CaBP2 with CaV1.3 inhibits CDI in HEK293 cells). The inactivation kinetics of CaV1.3 are also regulated by the subunit composition (Cui et al, 2007) along with the modulation via interaction partners and given the reconstitution here we do not find the currents very surprising.

(5) Is the difference in Ca2+-influx still significantly higher upon the removal of the maximum value measured in positive Syribbons spots (Figure 7, panel K)?

Yes, on removing the maximum value, the P value increases from 0.01 to 0.03 but remains statistically significant.

(6) In summary, although the approach pioneered by the authors is exciting and provides relevant results, there is a major concern regarding the interpretation of the modulation of Ca2+ channels.

We have now carefully rephrased our interpretation on the modulation of Ca^2+^ channels.

References

Brandt A (2005) Few CaV1.3 Channels Regulate the Exocytosis of a Synaptic Vesicle at the Hair Cell Ribbon Synapse. Journal of Neuroscience 25: 11577–11585

Cui G, Meyer AC, Calin-Jageman I, Neef J, Haeseleer F, Moser T & Lee A (2007) Ca2+-binding proteins tune Ca2+-feedback to Cav1. 3 channels in mouse auditory hair cells. The Journal of Physiology 585: 791–803

Davydova D, Marini C, King C, Klueva J, Bischof F, Romorini S, Montenegro-Venegas C, Heine M, Schneider R, Schröder MS, et al (2014) Bassoon specifically controls presynaptic P/Q-type Ca(2+) channels via RIM-binding protein. Neuron 82: 181–194

tom Dieck S, Altrock WD, Kessels MM, Qualmann B, Regus H, Brauner D, Fejtová A, Bracko O, Gundelfinger ED & Brandstätter JH (2005) Molecular dissection of the photoreceptor ribbon synapse: physical interaction of Bassoon and RIBEYE is essential for the assembly of the ribbon complex. J Cell Biol 168: 825–836

Frank T, Rutherford MA, Strenzke N, Neef A, Pangršič T, Khimich D, Fejtova A, Gundelfinger ED, Liberman MC, Harke B, et al (2010) Bassoon and the synaptic ribbon organize Ca2+ channels and vesicles to add release sites and promote refilling. Neuron 68: 724–738

Grabner CP & Moser T (2021) The mammalian rod synaptic ribbon is essential for Cav channel facilitation and ultrafast synaptic vesicle fusion. eLife 10: e63844

Hibino H, Pironkova R, Onwumere O, Vologodskaia M, Hudspeth AJ & Lesage F (2002) RIM - binding proteins (RBPs) couple Rab3 - interacting molecules (RIMs) to voltage - gated Ca2+ channels. Neuron 34: 411–423

Inoue E, Deguchi-Tawarada M, Takao-Rikitsu E, Inoue M, Kitajima I, Ohtsuka T & Takai Y (2006) ELKS, a protein structurally related to the active zone protein CAST, is involved in Ca2+-dependent exocytosis from PC12 cells. Genes to Cells 11: 659–672

Janigro D, Maccaferri G & Meldolesi J (1989) Calcium channels in undifferentiated PC12 rat pheochromocytoma cells. FEBS Letters 255: 398–400

Jean P, Morena DL de la, Michanski S, Tobón LMJ, Chakrabarti R, Picher MM, Neef J, Jung S, Gültas M, Maxeiner S, et al (2018) The synaptic ribbon is critical for sound encoding at high rates and with temporal precision. Elife 7: e29275

Koschak A, Reimer D, Huber I, Grabner M, Glossmann H, Engel J & Striessnig J (2001) alpha 1D (Cav1.3) subunits can form l-type Ca2+ channels activating at negative voltages. J Biol Chem 276: 22100–22106

Krinner S, Butola T, Jung S, Wichmann C & Moser T (2017) RIM-Binding Protein 2 Promotes a Large Number of CaV1.3 Ca2+-Channels and Contributes to Fast Synaptic Vesicle Replenishment at Hair Cell Active Zones. Front Cell Neurosci 11: 334

Liu H, Felix R, Gurnett CA, De Waard M, Witcher DR & Campbell KP (1996) Expression and Subunit Interaction of Voltage-Dependent Ca2+ Channels in PC12 Cells. J Neurosci 16: 7557–7565

Lv C, Stewart WJ, Akanyeti O, Frederick C, Zhu J, Santos-Sacchi J, Sheets L, Liao JC & Zenisek D (2016) Synaptic Ribbons Require Ribeye for Electron Density, Proper Synaptic Localization, and Recruitment of Calcium Channels. Cell Reports 15: 2784–2795

Matthews G & Fuchs P (2010) The diverse roles of ribbon synapses in sensory neurotransmission. Nat Rev Neurosci 11: 812–822

Maxeiner S, Luo F, Tan A, Schmitz F & Südhof TC (2016) How to make a synaptic ribbon: RIBEYE deletion abolishes ribbons in retinal synapses and disrupts neurotransmitter release. The EMBO Journal 35: 1098–1114

Michanski S, Kapoor R, Steyer AM, Möbius W, Früholz I, Ackermann F, Gültas M, Garner CC, Hamra FK, Neef J, et al (2023) Piccolino is required for ribbon architecture at cochlear inner hair cell synapses and for hearing. EMBO Rep 24: e56702

Michanski S, Smaluch K, Steyer AM, Chakrabarti R, Setz C, Oestreicher D, Fischer C, Möbius W, Moser T, Vogl C, et al (2019) Mapping developmental maturation of inner hair cell ribbon synapses in the apical mouse cochlea. PNAS 116: 6415–6424

Neef J, Urban NT, Ohn T-L, Frank T, Jean P, Hell SW, Willig KI & Moser T (2018) Quantitative optical nanophysiology of Ca2+ signaling at inner hair cell active zones. Nat Commun 9: 290

Park D, Wu Y, Lee S-E, Kim G, Jeong S, Milovanovic D, Camilli PD & Chang S (2021) Cooperative function of synaptophysin and synapsin in the generation of synaptic vesicle-like clusters in non-neuronal cells. Nat Commun 12

Picher MM, Gehrt A, Meese S, Ivanovic A, Predoehl F, Jung S, Schrauwen I, Dragonetti AG, Colombo R, Camp GV, et al (2017) Ca2+-binding protein 2 inhibits Ca2+-channel inactivation in mouse inner hair cells. PNAS 114: E1717–E1726

Robertis ED & Franchi CM (1956) Electron Microscope Observations on Synaptic Vesicles in Synapses of the Retinal Rods and Cones. J Biophys Biochem Cytol 2: 307–318

Roberts WM, Jacobs RA & Hudspeth AJ (1990) Colocalization of ion channels involved in frequency selectivity and synaptic transmission at presynaptic active zones of hair cells. J Neurosci 10: 3664–3684

Smith CA & Sjöstrand FS (1961) A synaptic structure in the hair cells of the guinea pig cochlea. Journal of Ultrastructure Research 5: 184–192

Wong AB, Rutherford MA, Gabrielaitis M, Pangršič T, Göttfert F, Frank T, Michanski S, Hell S, Wolf F, Wichmann C, et al (2014) Developmental refinement of hair cell synapses tightens the coupling of Ca2+ influx to exocytosis. EMBO J 33: 247–264

Zampini V, Johnson SL, Franz C, Lawrence ND, Münkner S, Engel J, Knipper M, Magistretti J, Masetto S & Marcotti W (2010) Elementary properties of CaV1.3 Ca(2+) channels expressed in mouse cochlear inner hair cells. J Physiol 588: 187–199